

# 3D surface properties of glacier penitentes over an ablation season, measured using a Microsoft Xbox Kinect.

L. I. Nicholson[1], M. Pętlicki[2,3], B. Partan[4], and S. MacDonell[3]

[1] *Institute of Atmospheric and Cryospheric Sciences, University of Innsbruck, Innsbruck, Austria*
[2] *Institute for Geophysics, Polish Academy of Sciences, ul. Księcia Janusza 64, 01-452 Warsaw, Poland*
[3] *Centro de Estudios Avanzados en Zonas Áridas (CEAZA), La Serena, Chile*
[4] *University of Maine, Orono, USA*

*Correspondance to:* L. I. Nicholson (lindsey.nicholson@uibk.ac.at)

**Abstract.** Penitentes are a common feature of snow and ice surfaces in the semi-arid Andes where very low humidity, in conjunction with persistently cold temperatures and sustained high solar radiation favour their development during the ablation season. As penitentes occur in arid, low-latitude basins where cryospheric water resources are relatively important to local water supply, and atmospheric water vapor is very low, there is potential value in understanding how penitentes might influence the runoff and atmospheric humidity.

The complex surface morphology of penitentes makes it difficult to measure the mass loss occurring within them because the (i) spatial distribution of surface lowering within a penitente field is very heterogeneous, and (ii) steep walls and sharp edges of the penitentes limit the line of sight view for surveying from fixed positions and (iii) penitentes themselves limit access for manual measurements. In this study, we solved these measurement problems by using a Microsoft Xbox Kinect sensor to generate small-scale digital surface models (DSMs) of small sample areas of snow and ice penitentes on Tapado Glacier in Chile (30°08'S; 69°55'W) between November 2013 and January 2014. The surfaces produced by the complete processing chain were within the error of standard terrestrial laser scanning techniques. However, in our study insufficient overlap between scanned sections that were mosaicked to cover the studied sites can result in three-dimensional positional errors of up to 0.3 m.

Mean surface lowering of the scanned areas was comparable to that derived from point sampling of penitentes at a minimum density of 5 m$^{-1}$ over a 5 m transverse profile. Over time the penitentes become fewer, wider, deeper and the distribution of slope angles becomes more skewed to steep faces. These morphological changes cannot be captured by the interval sampling by manual point measurements. Roughness was computed on the 3D surfaces by applying previously published geometrical formulae; one for a 3D surface and one for single profiles sampled from the surface. For each method a range of ways of defining the representative height required by these formulae was used, and the calculations were done both with and without using a zero displacement height offset to account for the likelihood of skimming air flow over the closely spaced penitentes. The computed roughness values are in the order of 0.01-0.10 m during the early part of the ablation season increasing to 0.10-0.50 m after the end of December, in line with the roughest values previously published for glacier ice. Both the 3D surface and profile methods of computing roughness are strongly dependent on wind direction. However, the two methods contradict each other in that the maximum roughness computed for the 3D surface coincides with airflow across the penitente lineation while maximum roughness computed for sampled profiles coincides with airflow along the penitente lineation. These findings highlight the importance of determining directional roughness and wind direction for strongly aligned surface features and also suggest more work is required to determine appropriate geometrical roughness formulae for linearized features.



## 1. Introduction

Penitentes are spikes of snow or ice ranging from a few centimetres up to several metres in height that can form during the ablation season on snowfields and glaciers under the right conditions. The conditions required for penitentes to form are dew point below 0°C, persistently low air temperatures and sustained strong solar insolation (Lliboutry, 1954). These conditions are frequently met at high elevation, low-latitude glaciers and snowfields, such as in the subtropical Andes (e.g. Hastenrath and Koci, 1981; Corripio and Purves, 2005; Winkler et al., 2009), where penitentes are widespread during the ablation season.

Observations show that penitente geometry is aligned with the arc of the sun across the sky and tilted toward the sun at local noon, highlighting the importance of solar radiation in penitente formation (Lliboutry, 1954; Hastenrath and Koci, 1981; Bergeron et al., 2006). Indeed, the alignment and restricted latitudinal range of penitentes (within 55° of the equator on horizontal surfaces) can be explained by solar-to-surface geometry alone (Cathles et al., 2014). The process of penitente growth involves geometric focusing of incident solar radiation by surface irregularities that causes depressions to receive more radiation than surrounding peaks (Amstutz, 1958; Corripio and Purves, 2005; Lhermitte et al., 2014; Claudin et al., 2015). Consequently, energy receipts and ablation are initially enhanced in the hollow due to multiple reflection of irradiance, and the surface irregularity becomes amplified. However for substantial penitente growth it is crucial that, at the tips of penitentes, ablation occurs by sublimation and the snow/ice temperature remains below the melting point, while in the troughs between penitentes, melting can occur once a more humid microclimate is established within the hollow (Lliboutry, 1954; Drewry, 1970; Claudin et al., 2015). Once the snow/ice in the hollows has reached the melting point, the spatial differentiation of ablation processes serves to further amplify the penitente relief as melting only requires approximately an eighth of the energy of sublimation to remove the same amount of ice.

The impact of penitentes on the surface energy balance and ablation of snow and ice is of interest in arid mountains catchments, where penitentes are widespread and meltwater can be a substantial contribution to local hydrological resources (Kaser et al., 2010). Previous studies have shown that penitentes alter the surface energy balance of snow and ice surfaces by reducing effective albedo by up to 40% compared to flat surfaces (Warren et al, 1998; Corripio and Purves, 2005; MacDonell et al., 2013; Cathles et al., 2014; Lhermitte et al., 2014) as well altering the partitioning of ablation between sublimation and melting (e.g. Lliboutry, 1998; Winkler et al., 2009; Sinclair and MacDonell, 2015). Thus, the presence of penitentes is expected to alter the rate of mass loss and meltwater production of snow and icefields during the ablation season, and, on the basis of the radiative balance it has been postulated that they will accelerate the snow and ice mass loss rates (Cathles et al., 2014). However, the development of penitentes on the surface will also alter the roughness properties in both space and time, but this, as well as its impact on the resultant turbulent fluxes is not quantified. The wind direction-dependence of surface roughness over linearized surface features has been previously observed in wind profile measurements over snow sastrugi, for which the derived aerodynamic roughness length varied from 1- 70 mm over 120° range of wind direction (Jackson and Carol, 1978). While penitentes are a relatively rare form of linearized surface feature in many glacierized environments, in contrast linear crevasses are widespread, and although the impact of wind direction on roughness and the resultant turbulent heat fluxes is generally not treated in glaciology, penitentes offer a unique test bed for investigating the significance of linearized features on effective surface roughness for various wind directions.

In general, the physical roughness of snow and ice surfaces are particularly prone to varying in space and time (e.g. Smeets et al., 1999; Brock et al., 2006; Fassnacht et al., 2009), it is desirable to be able to replace relatively logistically and technologically challenging methods of determining roughness parameters from atmospheric profile or eddy covariance measurements, with methods based on more readily measurable surface terrain properties (e.g. Kondo and Yamazawa, 1986; Munro, 1989; Andreas, 2011), or properties such as radar backscatter that can be derived from spaceborne instruments (e.g. Blumberg and Greeley, 1993). The most comprehensive surface of



methods to determine apparent aerodynamic properties from surface morphometry was carried out by Grimmond and Oke (1999) who tested several methods in urban environments, which are among the roughest surface conditions encountered in boundary layer atmospheric studies. The morphometric estimates of roughness properties were compared with those from aerodynamic methods from numerous field and laboratory studies. Many of the aerodynamic studies were found to be flawed, and the study demonstrates that, despite the considerable effort in obtaining such measurements, their reliability in complex and rough terrain is contested as the computations rely upon theory that is developed for flat homogenous terrain, and in general the aerodynamic results show a similar amount of spread as the various geometrical methods tested. Although, Grimmond and Oke (1999) consider that direct measurements of fluxes over complex terrain are most likely the 'best' way of determining surface properties, the difficulties of deploying the expensive and relatively delicate instruments over glacier surfaces makes a geometric determination even more appealing. However, in the case of penitentes, such studies are impeded by a scarcity of information on real penitente geometry.

Measurements of natural penitentes (e.g. Naruse and Leiva, 1997) are rare as they are generally found in relatively inaccessible areas and the complex surface relief poses a considerable impediment to movement and measurement, for example preventing the use of simplified automated tools such as photogrammetric determination of surface profile heights (e.g. Fassnacht et al., 2010; Manninen et al., 2012). Furthermore, accurately measuring the convoluted penitente surface is in itself a significant challenge, as it includes overhanging surfaces, which is problem for immobile line-of-sight surveying equipment. However, advances in close-range mobile depth-of-field sensors and efficient feature tacking software used in interactive computer gaming offer potentially useful tools that can be applied to generate small-scale digital surface models to resolve such problems in earth science (e.g. Mankoff et al., 2013). In this study sample plots of penitentes in snow on a glacier surface are scanned using a Microsoft Xbox Kinect sensor as a close-range mobile distance ranger to produce a series of small-scale digital surface models (DSMs). These surface models are used to perform (i) the first detailed examination of the geometry of penitentes and how they change over the course of the core ablation season; (ii) an examination of the geometrical roughness properties of penitentes and (iii) compare the volume changes computed from differencing the DSMs with the volume changes estimated from manual measurements of surface lowering within a penitente field. These measurements enable evaluation of how accurately simplified penitente surfaces used in theoretical modelling represent the true surfaces found in nature, improved parameterization of surface roughness in energy balance models applied to glacier and snowfields with penitentes, and the performance of energy balance models over penitente surfaces to be evaluated against mas loss derived from the measured surface changes.

## 2. Methods

### 2.1 Description of fieldsite

Tapado Glacier (30°08'S; 69°55'W) lies in the upper Elqui Valley of the semi-arid Andes of the Coquimbo Region of Chile (Figure 1). This glacier is relatively easily accessible and previous research indicates that the glacier surface develops penitentes every summer (Sinclair and MacDonell, 2015). Two separate study areas were analysed. Firstly, a test site was established at a patch of snow penitentes within a dry stream bed at 4243 m a.s.l. in the glacier foreland (Figure 1b). This site was used to (i) trial instrumental setups in order to optimize the field operation of the Kinect sensor, and (ii) compare the performance of the Kinect sensor against a Terrestrial Laser Scanning (TLS) system. This location was chosen due to the logistical difficulties of transporting the TLS to the glacier. Subsequently, two study plots were established at an elevation of 4774 m a.s.l. within the glacier ablation zone. These surfaces at these sites were measured repeatedly using the Xbox Kinect (see section 2.3) during the core ablation season between the end of November 2013 and the beginning of January 2014. An automatic weather station on a free-standing tripod was installed beside the two plots to provide meteorological context for the measurements.



The location and layout of the two sites is shown in Figure 1. Site A (5 m by 2 m) was measured four times, on 25
November, 11 December, 20 December and 3 January. Site B (2 m by 2 m, Figure 1c) was only measured on the last
three dates (Figure 1c). The corners of the study sites were marked with 2 m lengths of plastic plumbing piping
hammered vertically into the snow, or drilled into the ice. In order to locate the study sites in space and to provide a
common reference frame for each survey date, marker stake positions were measured using a Trimble 5700
differential GPS with Zephyr antenna on the 25th November, with a base station in the glacier foreland. On each
visit to the glacier, when possible, the stakes were hammered further into the snow and the resultant lowering of the
stake top was noted. The maximum standard deviations of the GPS stake positions were < 1.0 cm, 1.1 cm and
1.7 cm in easting, northing and elevation respectively, with combined XYZ standard deviation < 2.0 cm for all
stakes (Supplement A). Error on the manual measurements of height offsets of the marker stakes on subsequent
survey dates is conservatively estimated to be 2.0 cm. This results in total positional errors of the ground control
points at each scan date of between 2.3 and 2.7 cm depending on the stake.
**2.2 Terrestrial laser scanning**
Surface scans of snow penitentes at the test site were undertaken with both a terrestrial laser scanner (TLS) and the
Kinect sensor in order to compare the surface scans produced by the well-established TLS method and the relatively
new Kinect sensor application. The TLS system used was an Optech ILRIS-LR scanner, which is a long-range
terrestrial laser scanner especially suitable for surveying snow and ice surfaces thanks to a shorter wavelength laser
beam (1064 nm) than other models. This equipment surveys surface topography based on time-of-flight
measurement of a pulsed laser beam reflected to a given angle by a system of two rotating mirrors. It has a raw
range accuracy of 4 mm at 100 m distance, raw angular accuracy of 80 µrad, beam diameter of 27 mm at 100 m
distance and beam divergence of 250 µrad. The instrument was placed in five locations around the surveyed snow
patch and boulder, overlooking it from different directions. Positions of the TLS were measured with the Trimble
5700 differential GPS with Zephyr antenna in static mode. Seventeen point clouds were obtained with nominal
resolution of 0.11-0.75 cm. Resulting point clouds were corrected for atmospheric conditions (pressure, temperature
and humidity) and trimmed with ILRIS Parser software, aligned with Polyworks IMAlign software into a common
local coordinate system and georeferenced with differential GPS measurements using Polyworks IMInspect
software. The alignment error of the point clouds as estimated by software is 0.36-0.87 cm and comparison with
ground control points gives an error of 5.65 cm. Unfortunately, the scans of snow penitentes could not be carried out
with both the TLS and Kinect on the same day, so direct comparison of the TLS and Kinect scans is instead
performed on a reference boulder lying on the ground beside the test site, whose surface is assumed unchanged
between different scan dates. The TLS scan of the snow penitentes is presented as an example of the nature of the
DSM that can be obtained within a penitente field using a TLS (Figure 2).
**2.3 Kinect scans of surface change**
The Kinect sensor emits a repeated pattern of structured infra-red (IR) beams, and records the pattern distortion with
an onboard IR camera. The depth of field calculation is performed via a proprietary algorithm and a distance map is
the raw data output. Using the standard calibration the static raw depth field resolution of the Kinect is 1 mm and the
Kinect-measured distance at the center of the field of view is within 1% of the real distance (Mankoff et al., 2013),
implying an error of < 1.0 cm at the distance range of the penitente scans.
For its original gaming usage, the Kinect is in a fixed position and proprietary software uses feature tracking to track
the movements of players moving within the field of view of the Kinect. However, the inverse of this workflow can
also be applied wherein the Kinect sensor is moved interactively around a static surface or 3D body, using the same
feature tracking to compute the position of the sensor relative to the object and thereby allowing a point cloud
reconstruction of the object to be constructed. In this work we apply the second work flow sampling Kinect data
using the ReconstructMe™ 2.0 software package. In common with alternative reconstruction packages that are
compatible with the Kinect, ReconstructMe™ performs bilateral filtering on the output depth map frame and



174 converts the pixel version of each depth map frame to 3D coordinate maps of vertices and normals. An iterative
175 closest point (ICP) alignment algorithm is then applied frame by frame at three scales to repeatedly rotate and
176 translate the depth field to determine camera position and an aligned surface, giving weighted preference to portions
177 of the surface that are perpendicular to the line of sight. This software has the advantage of producing surface
178 meshes in real-time, so that the operator can visibly check the scan quality and coverage at the time of capture, but
179 the disadvantage that the raw point cloud is not saved and if the real-time tracking is lost a new scan sample must be
180 started.

181 The Xbox Kinect was connected via a 5m powered USB extension cord to an MSI GE60 gaming laptop, powered
182 using a 240V 600W inverter connected to the 12V 160Ah battery of the automatic weather station on the glacier.
183 Scans were carried out by two people; one moving the Kinect across the penitente field and the other monitoring the
184 quality of the surface being generated. The return IR signal of the Kinect is swamped by natural radiation in bright
185 conditions, and this is especially true over bright, rough snow and ice surfaces, which reflect the shortwave
186 radiation, and absorb or scatter much of the longwave radiation signal. To solve this, scanning was carried out at
187 twilight or just after nightfall. Sudden movements caused by the operator slipping or the snow compacting underfoot
188 can result in the ReconstructMe software losing its tracking of common reference points used to generate the
189 continuous surface mesh. Consequently, each study site was scanned in small sections and three to thirteen separate
190 surface meshes were used to cover the area of each study site.

191 **2.4 Mesh processing**

192 Freely available Meshlab software was used to initially align the surface meshes covering each study site using a
193 pairwise alignment procedure. The full mesh processing procedure is presented in Supplement B, and briefly
194 described here. Small surface components, unreferenced and duplicated vertices were removed from the meshes
195 using inbuilt filters. The Meshlab alignment algorithm was applied to objectively optimize the alignment and
196 compute the alignment error. This alignment procedure uses an ICP algorithm to iteratively align the component
197 meshes and distribute the alignment errors evenly across the mosaicked surface mesh. Alignment solutions
198 consistently had mean distributed error < 4 mm (Supplement B). The aligned meshes were flattened into a single
199 layer, remeshed using a Poisson filter and finally resampled to reduce the point density by setting a minimum vertex
200 spacing of 2.5mm.

201 The surface mesh for each scan date was georeferenced using the known coordinates of the base of the marker
202 stakes at the time of each scan because the upper portions of the stakes are often poorly represented in the scans due
203 to the fact that ReconstructMe™ does not handle symmetrical objects well. It proved difficult in some cases to
204 locate the surfaces in space such that the locations of all marker stakes were consistent with the ground control
205 points. This is most likely an artifact of a combination of (i) reduced mesh quality at the margins of the component
206 scans and (ii) insufficient overlap between some scan sections producing distortion within the mesh alignment. The
207 mismatch evident in the georeferencing step (Table 1) is much larger than the mesh alignment error (Supplement B).

208 To eliminate the marker stakes and any data gaps near the margins of the study areas, each surface mesh was sub-
209 sampled within the staked area. The sub-sampled area for site A is a 2.0 by 3.5 m horizontal area (7.00 $m^2$), and site
210 B is a 1.5 x 1.5 m horizontal area (2.25 $m^2$) shown in the examples in Figure 3. Mesh vertices and an index file of
211 the vertices comprising each face were exported from Meshlab for subsequent analysis in Matlab.

212 **2.5 Calculations of surface geometrical properties**

213 The geo2d and geo3d toolboxes (available from the Matlab File Exchange) were used in Matlab™ to compute the
214 triangle areas and normals from which the surface height distribution, aspect and dip of the sampled surface can be
215 determined, weighted by the triangle area as a function of the total surface area of all faces. Volume change between
216 surfaces was computed by projecting each triangle area onto a baselevel horizontal reference surface. Volumes for



upward-facing triangles were computed column-wise from these projected areas using the height coordinate of the
triangle centroid as the height dimension for each column. These were summed and volumes for overhanging
triangles, calculated in the same way as the up-ward facing volumes were subtracted to derive a total volume
between the reference surface and the scanned penitente surface. Successive volumes were subtracted to obtain the
volume change over each measurement interval.

**2.6 Calculations of geometric surface roughness**

The aerodynamic roughness length ($z_0$) is the distance above the surface at which a logarithmic windspeed profile
under neutral conditions would be extrapolated down through the surface layer and reach zero. Over taller roughness
elements the level of action of momentum transfer between the airflow and the surface roughness elements is
displaced upwards by a distance, termed the zero-plane displacement ($z_d$). Above particularly rough surfaces, a
roughness sub-layer is formed in the lowest part of the surface layer within which surface roughness elements create
a complex 3D flow that is almost chaotic. Where roughness elements are widely spaced, the separated flow over
obstacles reattaches to the surface before the subsequent obstacle is reached. More closely packed roughness
elements experience a wake interference regime, and in the most densely packed arrays of roughness elements
skimming flow occurs (Grimmond and Oke, 1999). At the top of the roughness sublayer individual wakes caused by
surface obstacles are smeared out and the flow is independent of horizontal position, and thus, observations at this
level represent the integrated surface rather than individual surface obstacles. This level is known as the blending
height ($z_r$). All these properties are dependent on the size and arrangement of surface roughness elements.
There are a number of formulations for deriving $z_0$ from geometrical measurements. For example, the simplest
approach is to take the standard deviation of the surface elevations as a measure of roughness (Thomsen et al.,
2015). In this work, the surface meshes were analysed for roughness on the basis of a widely-used relationship
established by Lettau (1969), initially developed for isolated, regular obstacles distributed over a plane:
$$z_0 = 0.5\,h\left(\frac{s}{S}\right) \qquad\qquad (1)$$

where $h$ is the height of the obstacles, $s$ is the upwind silhouette area of each obstacle and $S$ is the specific area
occupied by each roughness element obstacle, also referred to as its lot area. The roughness values computed using
Equation 1 over 3D snow surfaces has been shown to vary widely depending on the methods of surface interpolation
used (Fassnacht et al., 2014), due to the influence on interpolation method on the unit surface area occupied by each
roughness element. However in this work the high resolution meshes used can be expected to adequately capture the
surface properties as no extrapolation or interpolation procedure is needed. Isolated roughness elements of regular
geometry distributed over a horizontal plane are a poor analogy for the irregular surface topography of a penitente
field, and the applicability of this formulation over penitentes has not been established. Nevertheless, we apply the
analysis as an illustration of the nature of the results generated from such an approach over penitentes and hope that
future aerodynamic roughness lengths can be compared to these geometrically derived ones. Macdonald and others
(1998) state that for irregular obstacles $h$ can be replaced by average obstacle height, $s$ with the sum of all the
upwind silhouette areas, and $S$ with the total area covered by the obstacles. While the upwind silhouette area, and
indeed surface area in any direction, is relatively easily defined for each surface mesh area using trigonometry, it is
difficult to define individual roughness elements and their representative heights, due to the lack of an apparent base
level. Here we first detrend the surfaces to remove any general surface slope at the site, then compute the roughness
for the detrended 3D meshes assuming that the roughness elements cover the whole surface area (i.e $S$ = plot area),
and for four possible representations of average obstacle height ($h$) as follows: (i) the maximum range of the
detrended mesh; (ii) twice the standard deviation of the detrended surface mesh; (iii) mean mesh height above the
mesh minimum; and (iv) median mesh height above the minimum.



These data are computed for illustrative purposes only as it is reported that Equation 1 fails when the roughness
element density exceeds 20-30%, as is expected for penitente fields (Macdonald et al., 1998). High density
roughness elements means that they interfere with the airflow around each other, and the zero wind velocity level is
displaced upwards, and effective roughness is a result of the roughness elements above this zero velocity
displacement plane. The zero displacement height in this sense, gives an indication of the penetration depth of
effective turbulent mixing into the penitente field. Accordingly, we additionally present sample calculations of
three-dimensional roughness on the detrended surface meshes using three possible realizations of $z_d$, as $z_d$ is also
unknown in the case of the penitente fields being sampled. In the first case, $z_d$ is taken to be $h$, in the second 2/3 $h$,
which is a widely used standard in forests and other complex terrain applications (Brutseart, 1975), and in the third
1/3 $h$ for comparison, both computed for the four realizations of $h$ used as before. Equation 1, (for irregular
obstacles) is then applied to the roughness elements remaining above the plane of the general surface slope offset by
a distance $z_d$ above the minimum height of the surface mesh. The representative height $h$ for this portion of the mesh
exceeding the plane is taken to be the mean area-weighted height of all triangles above this plane, $s$ is the summed
frontal area of all mesh triangles above $z_d$ that face into the chosen wind direction and $S$ is the total horizontal area of
the surface components above $z_d$.
Munro (1989, 1990) modified the formula of Lettau (1969) to be applied to a single irregular surface cross-section
of length $X$, sampled perpendicular to the wind direction. This modified formulation is easier to work with on a
glacier where the roughness elements are irregular, closely spaced, and generally poor approximations of objects
distributed over a plane.  Instead of having to define an obstacle height above the plane, $h$ is replaced with an
effective height $h*$ expressed as twice the standard deviation from the standardized mean profile height; $s$ is replaced
with $h*X/2f$, in which $f$ is the number of profile sections that are above the mean elevation; and $S$ is replaced with
$(X/f)^2$. This approach approximates the surface elevation profile as rectangular elements of equal size, and has been
shown to give results within 12% of the silhouette area determined by integrating between true topographic minima
(Munro, 1989). Importantly, roughness values derived this way over snow, slush and ice surfaces show reasonable
agreement with roughness values derived from wind profiles (Brock et al., 2006). To investigate the nature of the
roughness computed this way for north-south and east-west impinging wind directions, cross profiles longer than
1.5 m at 0.1m intervals orientated E-W and N-S were extracted from each scanned surface. Cross-sections were
detrended to remove the influence of any general surface slope at the site, and roughness was computed following
the modifications of Munro for each detrended surface profile. Mean profile roughness for these two wind directions
are presented for each sampled surface.
**2.7 Manual measurements of surface change**
Traditional stake measurements of glacier surface lowering made at a single point are unreliable within the
inhomogeneous surface of a penitente field, as multiple measurements are required to characterize the complex
surface. One alternative is to measure surface lowering at intervals along a profile perpendicular to the main axis of
alignment of the penitentes. Such a reference was installed along the 5 m-long eastern margin of site A, between two
longer corner stakes drilled 3 m into the ice using a Kovacs hand drill. The distance between a levelled string and
the glacier surface was measured using a standard tape measure at 0.2 m intervals on 23 November. Subsequent
measurements, on the 12 and 21 December and on 4 January, were made at 0.1 m intervals. All measurements were
recorded to the nearest centimetre, and the error on each measurement is conservatively estimated to be 2.0 cm,
which is assumed to capture the error associated with the horizontal position of the measurements along the
reference frame and the vertical measurements of the distance to the surface beneath.





## 3. Results

### 3.1 Evaluation of the quality and suitability of penitente scans by TLS and Kinect

At the test site, the snow penitentes were well-developed and between 0.5 and 1.0 m in height (Figure 1b). TLS scans were made of these penitentes to illustrate the capabilities of this more conventional scanning system in capturing the penitente surfaces. TLS scans were taken from five different vantage points positioned above the penitentes. The penitente surface produced by the TLS had surface slope ranging between -30 and 90 degrees, indicating that overhanging surfaces within the penitente field are captured, however only 58% of the total surveyed horizontal area could be scanned as the deepest parts of the troughs were obscured from the view of TLS by the surrounding penitentes (Figure 2a). By comparison, the hand-held, mobile nature of the Kinect means that 100% of the surface of the penitente field can be captured as the field of view can be adjusted into limitless close-range positions. The long range of the TLS makes it easier to cover large areas in comparison to the close range Kinect sensor, but as only penitente tips are scanned the utility of this larger areal coverage is limited.

The Kinect scan of the reference boulder produced from three mosaicked meshes was aligned to that produced from TLS point clouds. The TLS scan was incomplete, with parts of the top and overhanging surfaces of the boulder missing due to being obscured from the TLS survey positions, while the Kinect scan achieved complete coverage of the boulder. The difference between the two aligned meshes where overlapping data existed was always < 2 cm (Figure 2b), which is well within the error of the georeferenced TLS surface model. Larger differences in Figure 2b, up to 5 cm, occur only where there are holes in one of the surfaces being compared.

It is difficult to formally assess the total error of the surfaces produced by the Kinect scans because the proprietary software, ReconstructMe™ and Poisson surface reconstruction in Meshlab, are all black box processing steps in the workflow. The mean alignment errors of the mesh mosaicking step in Meshlab is < 0.4 cm and quantifiable errors associated with the GPS positions, subsequent measurement of the stake bottom positions relative to the GPS positions are all < 2.0 cm. However, in this study the three-dimensional georeferencing error is large (Table 1) compared to the other sources and can be taken as a reasonable value for the error of the total process chain. Errors given on the seasonal mass, volume and surface changes are based on summing the squares of the mean elevation difference between the marker stakes and ground control points (GPCs) at each site on the first and last survey dates.

### 3.2 Meteorological conditions

During the study period one significant snowfall event occurred on the 8[th] December 2013, when the sonic ranger recorded an increase of surface height of 0.09 m over the course of the day, and temperature and incoming longwave radiation increase progressively (Table 2). The surface conditions of albedo and surface temperature are derived from radiation measurements that integrate the signal from a sample area beneath the instrument. Surface temperature was calculated from measured surface longwave emissions, assuming a surface longwave emissivity of 1. Over the study period, albedo decreases and derived surface temperature increases (Table 2). Thus over the course of the study, the atmospheric energy supply increases and the surface properties become gradually more conducive to melting. In the three measurement periods 22, 38 and 43% of hourly values of surface temperature exceed the melting point and the warming atmosphere is clearly expressed in the positive degree days of the three periods which are 3.7, 2.2 and 31.5 over the 16, 9 and 14 day-long periods respectively. The height change difference between hourly mean sensor to surface distance recorded by the AWS sonic ranger at midnight at the end of the survey days indicates lowering rates of 17, 37 and 56 mm day$^{-1}$ over the same measurement intervals, indicating that the increasing energy receipts translate into increasing rates of surface lowering at the AWS.



### 3.3 Areal scans of penitente surfaces


Surface lowering rates derived from the computed volume changes per unit area are 21, 41 and 70 mm day$^{-1}$ over
each interval at site A and 57 and 61 mm day$^{-1}$ over the last two intervals at site B. Surface lowering calculated as
the difference between successive hypsometric mean mesh elevation for each site were within a few millimetres of
the volume computations: 22, 38 and 69 mm day$^{-1}$ for the three measured intervals at site A, and 54 and 60 mm day$^{-1}$
for the last two intervals at site B. The total surface lowering over the whole available period computed by volume
change (hypsometric mean height change) was 1.68 (1.77) ± 0.11 m at site A and 1.37 (1.32) ± 0.38 m at site B.
Surface height changes recorded at site A over the same period as at site B were 1.35 (1.31) ± 0.21 m, indicating that
the values were repeatable across both sites. The volume loss was converted to mass loss on the basis of the mean
snow density of 426 kg m$^{-3}$ (with an assumed error of ± 5%) measured in a 1.10 m snow pit excavated on 22
November 2013 beside the weather station. Mass loss at site A computed from mesh volume changes (hypsometric
height changes) between 25 November and 3 January was 716 ± 58 (754 ± 59) kg m$^{-2}$, indicating an underestimation
of mass loss but that the two computation methods are within error of each other. Mass loss at site B from mesh
volume changes (hypsometric height changes) between 11 December and 3 January was 582 (562) ± 166 kg m$^{-2}$.
Measurements at site A over the same period give mass loss of 573 (558) ± 95 kg m$^{-2}$, so again, measurements at
both sites are within error of each other.
The morphometry of the sampled penitentes changed visibly over the measured intervals (Figures 3 and 4). The
strong east-west preferential orientation predicted from theory developed early and was maintained throughout study
period. The expression of this alignment is more convoluted in the stages of development studied here than the
parallel rows of penitentes used in model representations (Corripio and Purves, 2005; Lhermitte et al., 2014). Over
time the penitente troughs became fewer in number, but wider and deeper inkeeping with the increasing surface
relief evident in the manual measurements. This is reflected by increasing total surface area, with the penitente
surfaces at site A providing between 1.7 and 4.0 times the surface area of the horizontal equivalent area, and at site
B providing between 2.1 and 3.7 times the horizontal surface area equivalent (Figure 4 a & b). Snowfall during the
first measurement interval decreases the surface area over that interval. The surface relief expressed by the vertical
range of the mesh also increases through time, except when snowfall partially filled the developing penitentes,
reducing both the range of the surface and the general slope angle. Nevertheless, the morphometric properties of the
meshes broadly meet the properties of simplified surfaces. The largest part of the surface is facing southwards, and
the predominant angle generally steepens over time, though again this trend is reversed by snowfall (Figure 4 c & d).
From the onset of measurements the surface aspect distribution is strongly dominated by north and south facing
components and this becomes more pronounced in the latter measurements and the preferred orientation rotates
slightly over the course of the season (Figure 4 e & f).

### 3.4 Surface roughness assessments


Given that aerodynamic measurements to determine the most suitable representative height and zero displacement
level for penitentes are thus far unavailable, the approach taken here was to do an exploratory study and compute
geometric surface roughness values using various ways of expressing $h$ and $z_d$. As a consequence the results are
purely illustrative and while patterns can be drawn from them that have meaning for understanding the nature of the
computation, the applicability of these values in turbulent exchange calculations remains to be established. The
representative height, $h$, used in the calculations increases over time in all cases, and is bounded by the case taking $h$
to be the range of the detrended surfaces (maximum) and the case taking $h$ as twice the standard deviation of the
detrended surface (Figure 5). For clarity, the other two case values are not included in the plots shown here.
Differences within a single method between the two sites can reach as much as 0.2 m.
The application of Lettaus (1969) formula is considered to be invalid if the ratio of the frontal area to the planar area
of the obstacles exceeds 0.2 – 0.3, with 0.25 often being chosen as a single value. In all cases of the penitente
surfaces this ratio exceeds 0.2, and only 6% of cases computed at 10° intervals of bearing over all dates are below





0.3, and these are all early in the season, before the 20th December. Exceeding this threshold implies that the
obstacles are so closely packed that 'skimming' airflow will occur. Ignoring this issue, calculated $z_0$ values increase
with time and show a strong dependence on the impinging wind direction, with values peaking for wind directions
perpendicular to the alignment of the penitentes (Figure 6). Calculated $z_0$ ranges from $0.01 - 0.90$ m, depending on
the way in which the representative height is expressed, the time of year and the wind direction (Figure 7). However,
given the close spacing of the penitentes it seems appropriate to also explore what the calculated $z_0$ would be like
when applying a zero displacement height offset, although again, in the absence of validation data these numbers
can be only indicative of the pattern of roughness computed by these methods. Introducing the zero displacement
height reduces the maximum calculated roughness by about half, and also reduces the variability between different
representative heights (Figure 7), as a smaller $h$ value translates into a smaller $z_d$ so that the calculation is performed
on a larger portion of the mesh.
Surface roughness assessments on the basis of calculations following Munro's modification for single profile
measurements were applied to cross profiles longer than 1.5 m yielding 20 (6) profiles oriented N-S and 33 (7) E-
W at site A (B). Surface amplitude increases over time, and the amplitude of the N-S running cross profiles is
generally larger than the E-W running cross profiles, as illustrated in the example of site B (Figure 8). The profile-
computed roughness length increases monotonically over time at site B, but shows a reduction over the first period
at site A, associated with snowfall during this period. Both the range and relative increase in roughness over time is
larger for the N-S running profiles. The computed roughness at both sites is 4.3 to 6.8 times larger for airflow
impinging on the penitente field in an E-W direction than for airflow in the N-S direction. This is contrary to the
results computed on the full 3D mesh surface, but is understandable because this formulation relies on the amplitude
of the surface, which is generally larger in the N-S orientated cross profiles than the E-W running cross profiles.
Prevailing wind direction differs only slightly in each period with an increasing northwesterly component in the
second two periods compared to the first. This may be related to the occurrence of snow during the first period,
which can be expected to alter the thermally driven valley wind systems. Over the whole study period wind direction
is predominantly from the south-westerly sector, but swings through southerly to easterly thereby encompassing
both extreme wind angles used in the roughness calculations here (Figure 9). This indicates that the effective
roughness can be expected to differ significantly depending on the wind direction.
**3.5 Manual measurements of reference cross-profile**
Using data sampled at 0.2 m over 5.0 m, the maximum relief of the sampled penitente profile, defined as the range
of the maximum and minimum distance from the horizontal reference to the surface, increased through time, from
0.76, 0.83, 1.00 to 1.38 m on each measurement date. The standard deviation of the surface remained relatively
unchanged over time with values of 0.24, 0.26, 0.28 and 0.32 m at each measurement date. The difference in the
mean surface height measured at the ablation frame profile at site A indicates mean lowering rates of 13, 57 and 61
mm day$^{-1}$ over the three sampled intervals resulting in a total mean surface lowering of $1.61 \pm 0.14$ m between 23 of
November and 4 January. The manual measurements at the cross profile compare well to the aerially-averaged
lowering rates from the scanned surfaces, despite the fact that the manual measurements are only made in 2
dimensions, do not visually represent the complexity of the penitente surfaces, and individual points are sometimes
out of the range of error of the Kinect (Figure 10). The computed mass loss over the same period is $688 \pm 70$ kg m$^{-2}$,
which underestimates the value for site A derived from volume changes but is within error, even accounting for the
two extra days measurement interval.
Values of maximum elevation range and standard deviation along the profile, mean surface height compared to the
horizontal reference and mean lowering were computed from the manual measurements for available data at 0.1 (n =
52), 0.2 (n = 26), 0.4 (n = 14) and 1.0 m (n = 6) intervals to investigate the impact of sampling resolution. The
highest resolution sample was taken as a reference against which to evaluate the values from coarser resolution
sampling. Calculated surface relief differed from that measured at the highest resolution by maxima of 0.13, 0.29





and 0.41 m for 0.2, 0.4 and 1.0 m sampling intervals respectively. Mean measured surface height was within 0.03 m
of the highest resolution measurements at 0.2 m and 0.4 m intervals, and within 0.12 m at 1.0 m resolution. Mean
lowering rates at 0.1, 0.2 and 0.4 m sampling intervals were all within 3 mm day$^{-1}$ with the difference increasing to a
maximum of 12 mm day$^{-1}$ when the sampling resolution was decreased to 1.0 m. Decreasing the length of the
sampled profile down to 2 m alters the mean lowering rate by less than 5 mm day$^{-1}$ at sampling resolutions of 0.1,
0.2 and 0.4 m.
Probing of the snowdepth on 25 November indicated a mean snow depth of 1.83 m (standard deviation 0.56 m).
The underlying ice surface identified by the snow probing, does not appear to be influencing the structure of the
snow penitentes developing in the current season. However, it is difficult to draw a firm conclusion based on
measurements at only 0.2 m spacing, particularly as, while the surface of the penitentes was still snow on the 3
January, in several instances the surface had lowered below the level of the ice interface indicated by the initial
probing.

## 4. Discussion

### 4.1 Methods of measuring change of rough glacier surface elements

The test site for scanning penitentes with a TLS was chosen as it provided the most optimal viewing angles possible
from scanning positions, as the penitentes lay in a river bed and scanning positions could be established on the
surrounding river banks to look down into the penitente field. Nevertheless, the terrestrial laser scanning could only
capture the tips rather than the whole surface of the penitentes and, as ablation is at its maximum in the troughs, TLS
data is therefore not able to determine the true volume change ongoing in penitentes. The coverage would be
increased if a higher viewing angle could be achieved, but the steep, dense nature of penitente fields makes it
difficult to imagine where sufficient suitable locations can be found surrounding glaciers or snowfields with
penitentes. In contrast, the mobile Kinect sensor can be moved across the complex relief of the penitente field to
make a complete surface model. Although it is in principle possible to capture a large area with the ReconstructMe
software used here, and it offers the advantage of providing real time feedback on the mesh coverage, it proved
difficult to capture the study sites in a single scan given (i) the reduced signal range of the sensor over snow and ice
(Mankoff et al., 2013), and (ii) the difficulty of moving around the penitente field. As a result, partial scans were
obtained, with the disadvantage that subsequently combining these introduces a substantial degree of additional error
associated with alignment if the component scans were not of high quality at the margins, or did not overlap
adjacent scan areas sufficiently. A combination of these two techniques might allow the extrapolation of small-scale
geometry changes and volume loss determined from a Kinect surface scan to be extrapolated usefully to the glacier
or snowfield scale.
Despite not visually capturing the complex surface properties of the penitentes, manual measurements of surface
height change in a penitente field along a profile cross-cutting the penitentes are robust for determining mean
surface lowering rates, and show good agreement to the volume changes computed from differencing the digital
surface models scanned in detail using a Kinect. Thus, the detailed surface geometry need not be known in order to
reasonably calculate the total volume loss over time within penitente fields. Comparison of the manual sampling at
different intervals suggest that five samples per meter is adequate to characterize penitentes. Over the 39 days of the
study, the mass loss calculated from 26 points spaced at 0.2 m intervals along a 5 m profile crosscutting the
penitentes differed from that calculated from volume change computed on surface meshes consisting of over 1.3
million points and covering an area of 7 m$^2$ by only 28 kg m$^{-2}$. Although this difference was within the error of the
two measurement types, the seasonal difference, assuming that this difference applies to a whole ablation season of
120 days would be 86 kg m$^{-2}$, and applied to the whole glacier (3.6 km$^2$) would amount to an underestimate of mass
loss over an ablation season of 0.3 gigatonnes. As a side note, the probing of snowdepth carried out as part of this
study highlights the difficulty in identifying the underlying ice surface, or summer ablation surface, in this way





within a penitente field, suggesting that a single location must be sampled very densely to obtain a characteristic
snowdepth in this way.

**4.2 Penitente morphometry and change in time**

The manual measurements at 0.2 m intervals are adequate to determine the mean surface lowering within a penitente
field, giving confidence to this type of simplified measurement on seasonal timescales. However, the interval
measurements cannot capture the surface morphometry, or how it changes in time.
At all times the penitente surface represents a much larger total surface area than the equivalent non-penitente
surface. Over time the surface relief, and slope angle, increases as the penitentes deepen, unless a snowfall event
occurs to partially fill the troughs, which also reduced the mean surface slope. The control of solar radiation on
penitente morphology means that the vast majority of the surface consistently dips steeply to the north and south at
all stages of development. This means that the angle of incidence of direct solar radiation is reduced, decreasing
both the intensity of the solar beam and the proportion of it that is absorbed. Although these effects are counteracted
by multiple reflections of solar radiation within the penitente (Corripio and Purves, 2005; Lhermitte et al., 2014;
Claudin et al., 2015) modeled mean net shortwave in an example penitente field at the summer solstice at 33°S is
about half of that of a level surface (Corripio and Purves, 2005). However, given the larger surface area of the
penitente field compared to a flat surface, the total absorbed shortwave is a third higher in the modeled penitentes.
At Tapado Glacier, penitentes are initially overhanging to the north, and the southfacing sides are convex compared
to the northfacing overhanging faces. Over the season the penitentes become more upright as the noon solar angle
gets higher. Idealized modelling based on measurements at Tapado Glacier, shows that concave and convex slopes,
as well as penitente size have been shown to impact the apparent albedo as measured by ground and satellite sensors
(Lhermitte, et el., 2014), and there may be some value in assessing the impact of these morphometry changes on
albedo over time. For the idealized penitente surface at 33°S during summer solstice case, modeled increase in net
shortwave radiation over penitentes is not compensated by modelled changes in net longwave radiation, meaning
that the excess energy receipts must be compensated by either turbulent energy fluxes or consumption of energy by
melting (Corripio and Purves, 2005).
In the context of the numerical theory of Claudin and others (2015), progressive widening of the penitente spacing,
as observed at both site A and B, is indicative of changes in the atmospheric level at which water vapor content is
unaffected by the vapor flux from the penitente surface. Simultaneous field or laboratory measurements of penitente
spacing evolution and vapor fluxes above the surface would be required to solidly confirm this, but the field
measurements provided here can be used as an indication of the level to which vapor flux from the surface is
influencing the boundary layer vapor content.

**4.3 Surface roughness**

In this work a single, simple geometric relationship (Lettau 1969) was investigated because a profile-based version
of this formulation has previously been tested against aerodynamic measurements over glacier surfaces (Munro,
1989, 1990; Brock et al., 2006). Certainly other relationships could be explored in the context of linearized glacier
features, but given the wide spread of values produced in previous comparisons such an analysis might be of limited
value in the absence of simultaneous aerodynamical investigations (Grimmond and Oke, 1999). Furthermore, the
results of Grimmond and Oke (1999) indicate that for the cities sampled, the Lettau method gives $z_0$ values that are
in the middle of the range of all the methods. The analysis of geometric computations of roughness properties in
Grimmond and Oke (1999) highlight the importance of correctly determining $z_d$, and limited sensitivity analyses
show the computed $z_d$ and $z_0$ to be strongly dependent on the dimensions of the obstacles. Lettau's (1969) formula,
which does not account for $z_d$, overestimates roughness for densely packed obstacles, but this overestimation does
not compensate sufficiently to reproduce values of $z_d + z_0$ produced for densely packed obstacles from formulations
that include $z_d$ in the computation of $z_0$. This means that Lettaus formula is expected to estimate the zero velocity



point of a logarithmic wind profile to be lower than formulations that include $z_d$ in their computation of $z_0$. In this
work however we computed $z_d$ in a separate preceding step to explore the impact of $z_d$ on the computed $z_0$.
As penitentes fields present very densely packed roughness elements, the frontal area of the surface tends to be large
compared to the ground area, and the limits of the ratio of frontal to planar area found in this study implies that
skimming flow is almost always occurring over penitente fields, such that turbulent airflow in the overlying
atmosphere does not penetrate to the full depth of the penitente fields. This is in agreement with the theory of
formation and growth of penitentes in which the development and preservation of a humid microclimate within the
penitente hollows is required to facilitate differential ablation between the trough and tip of the penitente. As the
spacing between the penitentes also increases over the ablation season the features become less densely packed over
time, although the available data are insufficient to determine if the spacing increases sufficiently by the end of the
season to comply with the applicable limits of the roughness calculation used here.
Application of geometrical roughness equations is made more problematic in penitente fields as it is not clear how
an appropriate representative obstacle height should be expressed, nor how to define the zero displacement level
during presumed skimming flow. Roughness calculated using a range of possible representations of these properties
point towards roughness values in the order of 0.01-0.10 m during the early part of the ablation season and 0.10-
0.50 m after the end of December. These values are in line with the roughest values previously published for glacier
ice (Smeets et al., 1999; Obleitner, 2000). The topographic analysis clearly shows that in the absence of intervening
snowfall events, this roughness increase is related to the deepening of the penitentes over time and an increase of the
surface amplitude. The patterns of the computed roughness properties is consistent between the two neighbouring
sites, but individual values can differ, suggesting that local relief varies substantially and sampling a larger area
would be beneficial in order to capture mean properties.
The strong alignment of penitentes means that roughness calculated on the 3D surface meshes is higher for wind
impinging in a north-south direction as the large faces of the penitentes form the frontal area in this case. In contrast,
if roughness is computed for individual profiles extracted from the mesh to mimic manual transect measurements in
the field, roughness is between 3 and 6 times larger for air flow along the penitente lineation (E-W) than it is across
the lineation (N-S). While clearly highlighting that the surface roughness of the strongly aligned penitente fields is
dependent on wind direction, this contradiction poses a conundrum as neither approach has been specifically
evaluated against independent surface roughness derived from atmospheric profile measurements over penitentes.
Consequently, although surface roughness calculations on the basis of profile geometry have been evaluated against
aerodynamic roughness over rough ice surfaces, the available data is insufficient to distinguish which pattern is
more appropriate for calculating turbulent fluxes over penitentes. It principle it sounds reasonable to expect airflow
across the penitente lineation to maximize turbulence as the penitentes present a large surface area to the wind, yet,
if skimming flow is established, with the result that only the tips of the penitentes are determining the structure of
the turbulence then roughness in this direction would be strongly reduced, and perhaps even be less than for air flow
along the penitente lineation, for which the smaller frontal area reduces the likelihood of skimming flow. Further
investigation of this in order to quantify the impact of penitentes on turbulent fluxes for various airflow patterns
requires measurement of turbulent fluxes using eddy covariance or atmospheric profile methods, which would
demonstrate the nature of the directional roughness and establish the impact of penitentes on turbulent energy fluxes
for different wind directions. Such measurements would be best implemented in a manner which can sample all
wind directions equally, and eddy covariance systems for which analysis is limited to a sector of airflow centred
around the prevailing airflow source, might not be able to capture the nature of the directional dependence correctly.
In this study we did not explicitly compute the blending height as available formulae are dependent upon $z_0$ and $z_d$.
Estimates of the blending height independently from $z_0$ and $z_d$ have been suggested to be 2.5 - 4.5 times $h$, as twice
the mean element spacing, or as combination of the height and spacing (see examples within Grimmond and Oke,
1999). Given that only atmospheric measurements above the blending height give representations of integrated
surface fluxes and conditions, the first approach would imply that aerodynamical or flux measurements over



penitentes would have to be carried out at some height above the surface to capture mean surface properties rather than the effects of individual roughness elements. The mathematical model of Claudin and others (2015) indicates that the level at which the vapour flux does not is constant in horizontal space, and therefore is the product of mean surface properties, is related to the spacing of the penitentes. Taking this to be representative of the blending height would imply that a formulation for the blending height might be possible on the spacing of penitentes alone, and that this in turn might contain useful data for understanding the structure and efficiency of turbulence above penitentes. However, exploring these ideas requires information from meteorological measurements as well as the geometrical information offered in this paper.

**5. Conclusion**

Surface scanning technology and software is an area of rapid development, and a number of potentially superior alternative set-ups and data capture sensors and software is now available. This study demonstrates that the Microsoft Kinect sensor can work successfully at close range over rough snow and ice surfaces under low light conditions, and generate useful data for assessing the geometry of complex terrain and surface roughness properties.

The data collected offers the first detailed study of how the geometry of penitentes evolves through time, highlighting the rate of change of surface properties over an ablation season that can serve as a guideline for parameterizing surface properties required for energy and mass balance modelling of penitente surfaces. The measurements confirm that even relatively crude manual measurements of penitente surface lowering are adequate for quantifying the seasonal mass loss.

Aerodynamical roughness properties and related metrics over very rough surfaces remain poorly quantified and both geometric and meteorological determinations of these values show a wide spread; consequently it remains unclear what the best methods to use are or what values modellers would be best to use (Grimmond and Oke, 1999). In this context penitentes and further study of them offers a useful opportunity as (a) their morphometric evolution over time allows various geometries to be evaluated by instrumenting and scanning a single site, and (b) they offer a bridge between wind tunnel and urban field experimentation of turbulence and roughness over extreme terrain. Although validity of surface roughness calculations based on surface geometry remains to be established for penitentes, this study highlights that (i) skimming flow is expected to persist over penitentes field, but is more likely under wind directions perpendicular to the penitente alignment; (ii) $z_d$ is certainly greater than zero, and while the depth of penetration of surface layer turbulence into a penitente field is not clearly established it is likely to evolve with the developing penitentes, and values of $z_d \sim 2/3h$ give results that are theoretically reasonable in the framework outlined by Grimmond and Oke (1999); (iii) the two methods of geometric computation of surface roughness applied here give conflicting results as to whether the effective surface roughness of penitentes is greater for airflow along or across the penitente lineation and (iv) more complete understanding of the impact of penitentes on the turbulent structure, its evolution in time, and its directional dependency, would require atmospheric measurements with no directional bias.

Potential future applications and analyses of the surfaces generated in this study include (i) using surface properties and roughness values as a guide for input into surface energy balance models; (ii) assessing the performance of models against the measured volume loss over time and (iii) evaluating how well simplified representations of penitente surfaces used in small scale radiation models and turbulence models capture the real-world complexity. Such studies would help establish the nature of the likely micro-climatic distribution of the surface energy balance within a real penitente field, and as a result the impact of penitentes on runoff and exchange of water vapour with the atmosphere.



**Author contributions.** L.N. designed the study. Fieldwork was carried out by L.N. and B.P. with M.P. providing the TLS. TLS and AWS equipment was provided by S.M. through collaboration with CEAZA. The data was analysed by L.N. and M.P. and L.N. prepared the manuscript and figures.

**Acknowledgements.** Fieldwork for this study was funded by a National Geographic Waitt Grant. L.N. was supported by an Austrian Science Fund Elise Richter Grant (V309). M. P. was supported within statutory activities No 3841/E-41/S/2015 of the Ministry of Science and Higher Education of Poland. International cooperation was supported by the Centre for Polar Studies from the funds of the Polish Leading National Research Centre (KNOW) in Earth Sciences (2014–18). Thanks are also due to Mathias Rotach for reading the paper prior to submission.

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

**Supplementary material**
- A: GPS position of ground control points at each glacier site
- B: Mesh surface components and processing steps used for Kinect surface scans
- C: Kinect surface meshes for both sites on all dates as .PLY files [sX_DDMM.PLY]
- D: 3D viewer files of surfaces at site B can be seen at:
https://sketchfab.com/LindseyNicholson/folders/penitentes-on-glaciar-tapado-chile



*Table 1: Maximum absolute georeferencing error at each marker stake for site A and B, relative to the standard deviation of the differential GPS measurement.*

|       | ΔX [mm] | ΔY [mm] | ΔZ [mm] | ΔXY [mm] | ΔXYZ [mm] | *dGPS XYZ standard deviation* [mm] |
|-------|---------|---------|---------|----------|-----------|-----------------------------------|
| **A-1** | 63 | 25 | 38 | 68 | 77 | *17* |
| **A-2** | 214 | 118 | 259 | 233 | 312 | *15* |
| **A-3** | 14 | 57 | 53 | 57 | 62 | *14* |
| **A-4** | 23 | 29 | 61 | 33 | 69 | *16* |
| **A-5** | 54 | 32 | 128 | 56 | 139 | *18* |
| **B-1** | 59 | 46 | 19 | 75 | 77 | *16* |
| **B-2** | 121 | 11 | 102 | 164 | 193 | *17* |
| **B-3** | 11 | 48 | 2 | 49 | 49 | *12* |
| **B-4** | 85 | 37 | 34 | 85 | 92 | *12* |



Table 2: Mean meteorological conditions during the measurement intervals: incoming shortwave (SW in), albedo (α), incoming longwave (LW in), windspeed (u), wind direction (dir), surface temperature computed from measured outgoing longwave radiation (T surface), air temperature (T air), relative humidity (RH), air pressure (P) and the distance between the sonic ranger and the glacier surface (dist).

| | SW in | α | LW in | u | dir | T surface | T air | RH | P | dist |
|---|---|---|---|---|---|---|---|---|---|---|
| | [W m$^{-2}$] | [-] | [W m$^{-2}$] | [m s$^{-2}$] | [°] | [°C] | [°C] | [%] | [hPa] | [m] |
| sensor | Kipp and Zonen CNR1 | | | Young 05103 | | CNR1 | Vaisala HMP45 | | Setra 278 | SR50 |
| 26/12 - 11/12 | 413 | 0.54 | 205 | 3.0 | 170 | -5.3 | -2.7 | 32.5 | 442 | 1.62 |
| 12/12 - 20/12 | 441 | 0.48 | 212 | 2.8 | 214 | -2.9 | -0.8 | 41.4 | 448 | 1.96 |
| 21/12 - 03/01 | 426 | 0.41 | 224 | 3.1 | 217 | -1.4 | 1.9 | 39.5 | 456 | 2.56 |





*Table 3: Surface roughness (z0) computed according to Munro (1989) on detrended profiles longer than 1.5 m, extracted at 0.10 m intervals from the Kinect surface meshes at site A and B for E-W impinging wind and N-S impinging wind. The number of profiles used for each wind direction is given in parenthesis. The likely displacement of the zero velocity plane (d_top ± standard deviation), was computed as the mean of 2/3h for all profiles and expressed as a distance from the top of the penitentes. The range of the detrended 3D mesh (3D range) provides a reference for the penetration depth of turbulence.*

| | site A | | | | | | site B | | | | | |
| | z0 E-W (20) | | | z0 N-S (33) | | | z0 E-W (6) | | | z0 N-S (7) | | |
| | mean | max | min | mean | max | min | mean | max | min | mean | max | min |
|---|---|---|---|---|---|---|---|---|---|---|---|---|
| **25-Nov** | 45 | 111 | 11 | 8 | 19 | 3 | | | | | | |
| **11-Dec** | 33 | 68 | 12 | 6 | 13 | 2 | 28 | 41 | 22 | 6 | 9 | 1 |
| **20-Dec** | 70 | 146 | 57 | 25 | 67 | 7 | 122 | 156 | 84 | 22 | 47 | 14 |
| **03-Jan** | 136 | 211 | 71 | 45 | 136 | 11 | 133 | 186 | 101 | 21 | 30 | 12 |

| | 3D range [m] | d_top +/- std [m] | | 3D range [m] | d_top +/- std [m] | | 3D range [m] | d_top +/- std [m] | | 3D range [m] | d_top +/- std [m] | |
|---|---|---|---|---|---|---|---|---|---|---|---|---|
| **25-Nov** | 0.41 | 0.27 | 0.06 | 0.41 | 0.34 | 0.02 | | | | | | |
| **11-Dec** | 0.48 | 0.33 | 0.05 | 0.48 | 0.41 | 0.01 | 0.58 | 0.45 | 0.02 | 0.58 | 0.51 | 0.02 |
| **20-Dec** | 0.76 | 0.58 | 0.03 | 0.76 | 0.61 | 0.04 | 0.98 | 0.76 | 0.02 | 0.98 | 0.84 | 0.04 |
| **03-Jan** | 1.07 | 0.79 | 0.03 | 1.07 | 0.86 | 0.05 | 1.14 | 0.86 | 0.03 | 1.14 | 0.98 | 0.02 |



Figure 1: Map of Tapado Glacier in the Elqui catchment of the Coquimbo Region of Chile, showing the location of the measured sites and insets of (a) the glacier site layout; (b) the test site highlighting the boulder (*) at which the Kinect scans were compared against TLS, and (c) an example photograph of glacier site B at the time of installation.

Figure 2: (a) Oblique view of the TLS derived DSM of the test site highlights the patchy coverage of the penitentes. (b) Absolute differences between DSMs of the sample boulder produced using TLS and Kinect.

Figure 3: Shaded meshes-S orientated DSMs for the 1.5m x 1.5m subsample at glacier site B on (a) 12.12.2013 (b) 20.12.2013 and (c) 03.01.201 obtained using the Kinect.

Figure 4: Summary of the DSM properties through time at site A (left) and B (right)..(a) and (b) Surface height distribution as a percentage of total surface area, in local coordinates [m] relative to the position of the northern end of ablation frame. Inset tables show weighted mean mesh elevation, range, surface area and surface area as a function of the horizontal area of the sampled site. (c) and (d) Distribution of surface angles as a percentage of total surface area. (e and f) Aspect distribution as a percentage of total surface area.

Figure 5: Representative surface heights computed on detrended surface meshes for site A (solid) and site B (open) over time where h1-h4 refer to representative surface heights computed as range (1), twice the standard deviation (2), area weighted mean height above the minimum (3), and area weighted median above the minimum mesh height (4).

Figure 6: 3D $z_0$ computed for 10° aspect intervals for all detrended DSMs highlighting peak roughness occurs in N-S airflow. Maximum values take h to be the detrended mesh elevation range, and minimum values take h to be twice the standard deviation of the detrended mesh.

Figure 7: Comparison of three-dimensional surface roughness through time, indicating the range of $z_0$ computed for all incident wind angles (at 10° intervals). Upper panels show the roughness with no zero level displacement and lower panels show values with a zero displacement offset d1 = h; d1 = 2/3h and d3 = 1/3h.As before, h1- h4 refer to representative surface heights computed as range, twice the standard deviation, area weighted mean height above the minimum, and area weighted median above the minimum mesh height respectively.

Figure 8: Examples of (a) N-S, and (b) E-W orientated cross sections sampled at 0.1 m intervals in local coordinates at site B from which effective surface roughness properties were computed using the methods of Munro (1989, 1999).

Figure 9: Wind rose for the whole study period (26 Nov 2013 – 3 Jan 2014).

Figure 10: Comparison of surface height through time extracted from the Kinect scan and measured manually along the horizontal reference. Error ranges on the Kinect cross profiles are given by a linear interpolation of total positional error between the bounding stakes. Solid black triangles indicate locations where snowdepth exceeded the length of the 3 m probe.









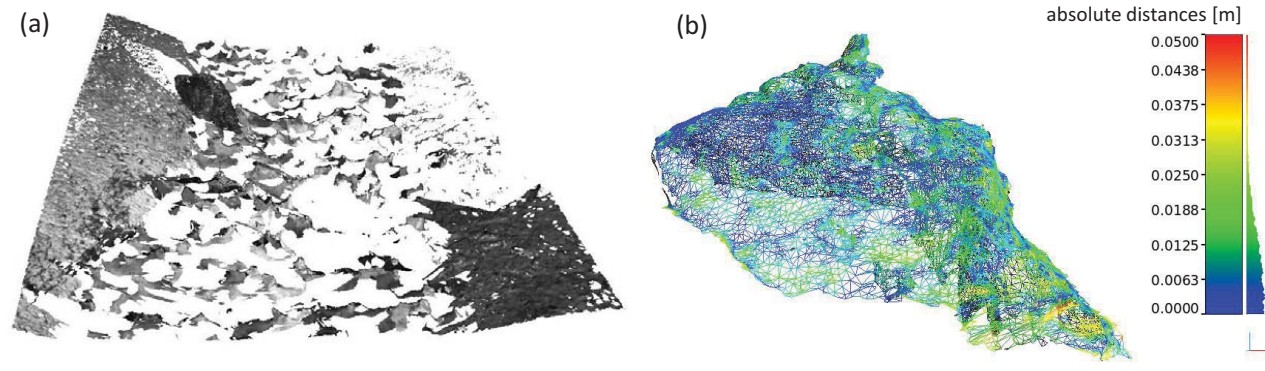



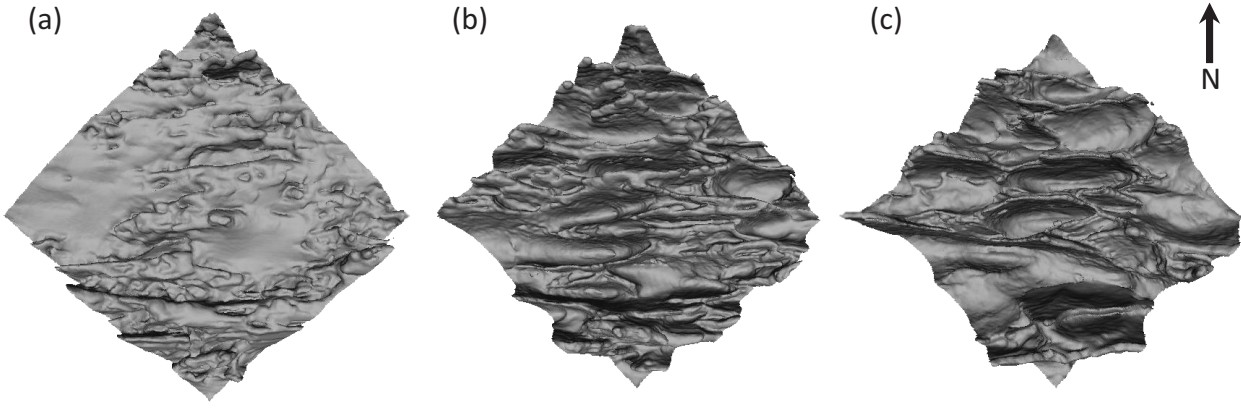





(a)

| site A | 25-Nov | 11-Dec | 20-Dec | 03-Jan |
|---|---|---|---|---|
| mean height [m] | -1.00 | -1.35 | -1.69 | -2.66 |
| height range [m] | 1.14 | 0.86 | 1.22 | 1.60 |
| surface [m²] | 15.04 | 13.13 | 12.13 | 27.95 |
| surface/area | 2.15 | 1.88 | 1.73 | 3.99 |

(b)

| site B | 25-Nov | 11-Dec | 20-Dec | 03-Jan |
|---|---|---|---|---|
| mean height [m] | | 0.19 | -0.3 | -1.13 |
| height range [m] | | 0.61 | 0.99 | 1.11 |
| surface [m²] | | 4.73 | 8.42 | 7.79 |
| surface/area | | 2.10 | 3.74 | 3.46 |

(c) (d) (e) (f)



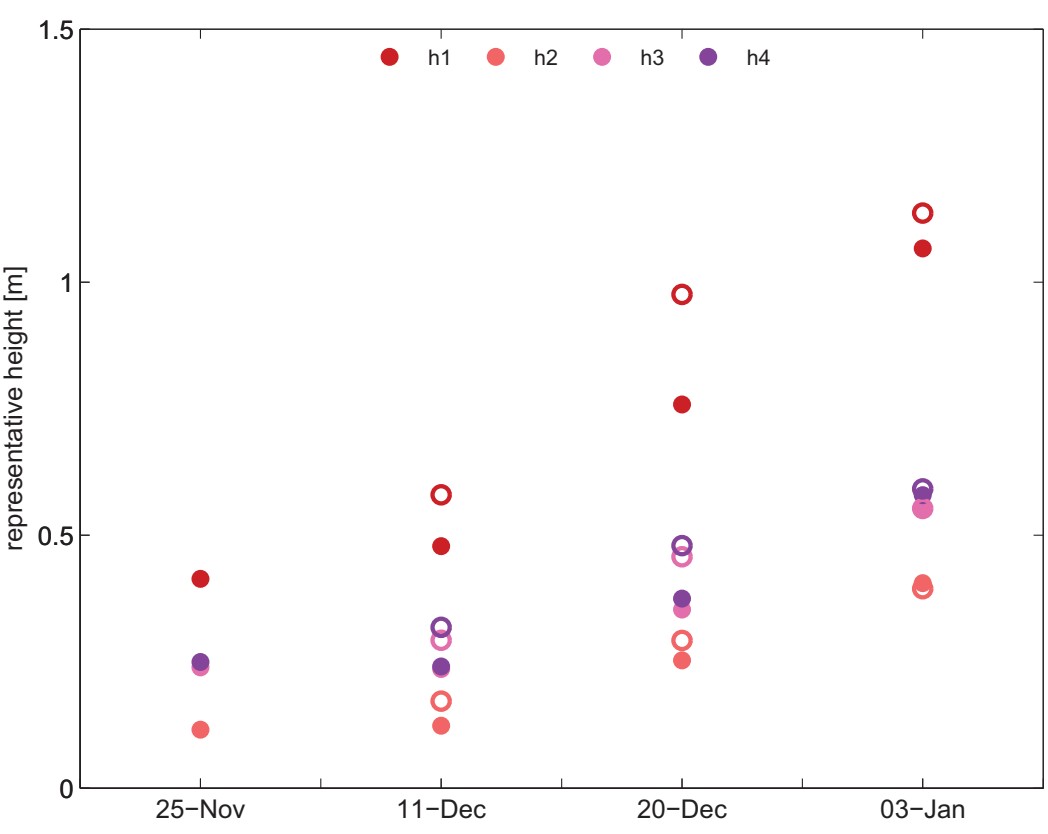





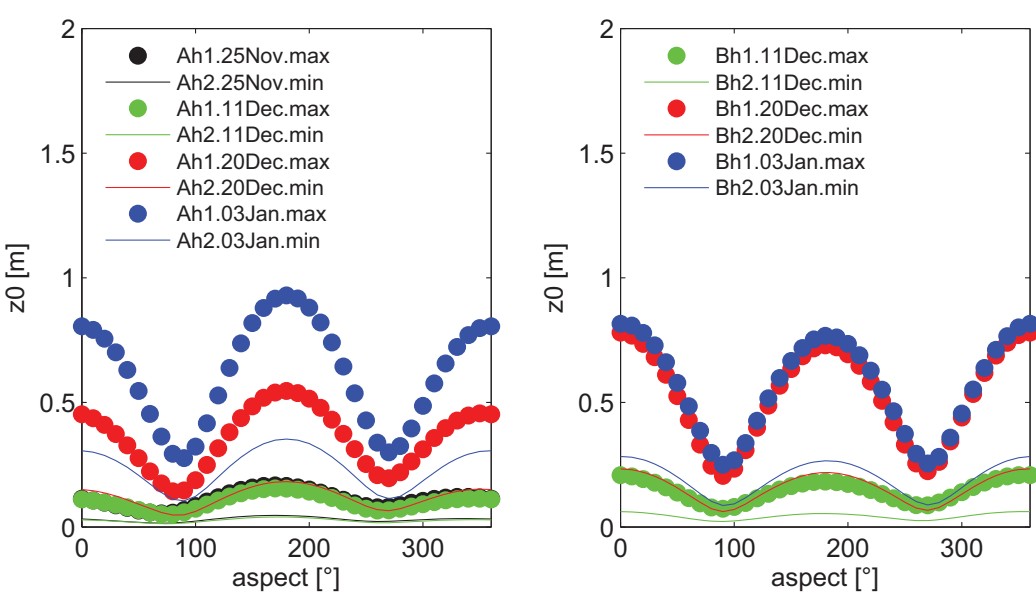





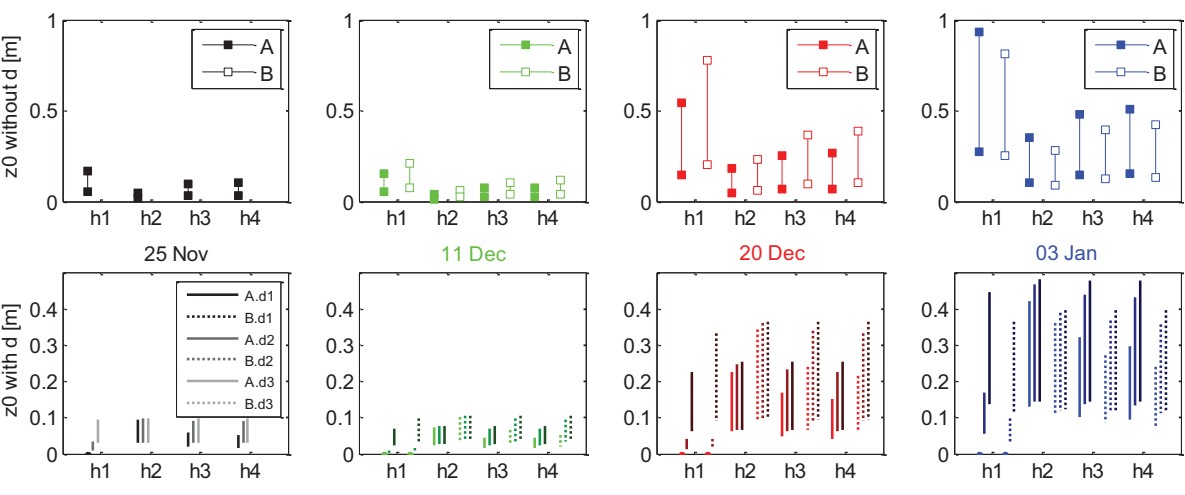



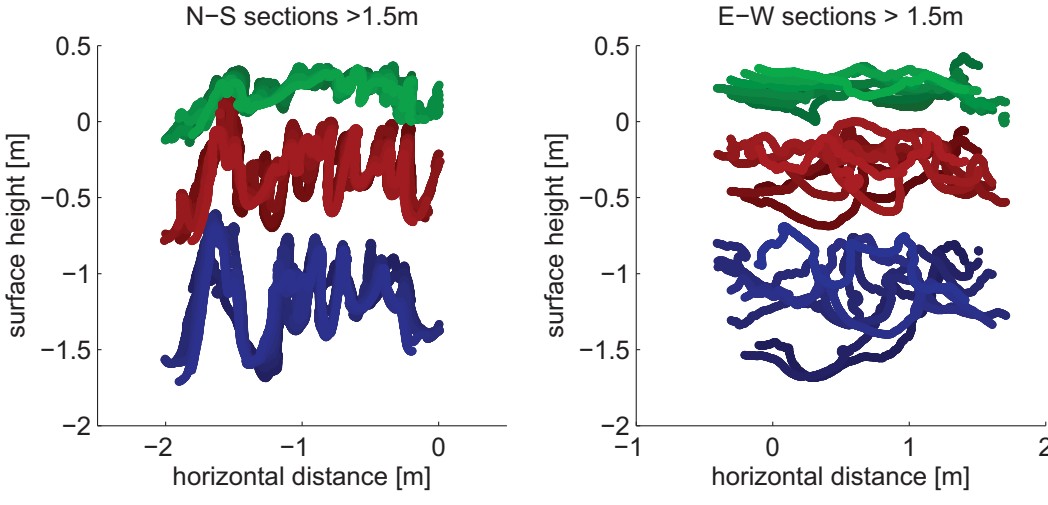





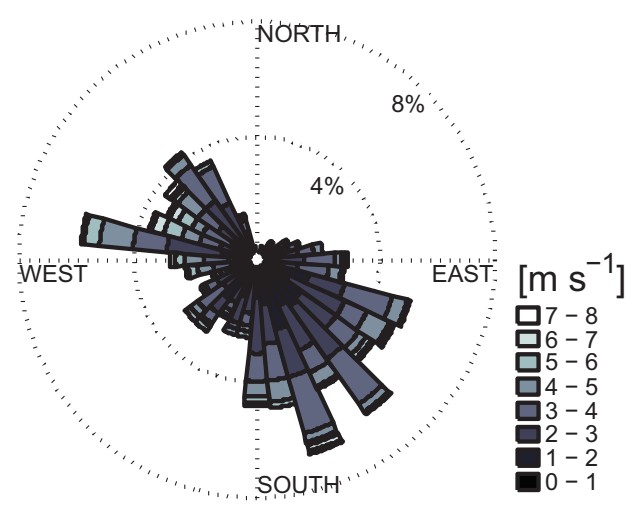





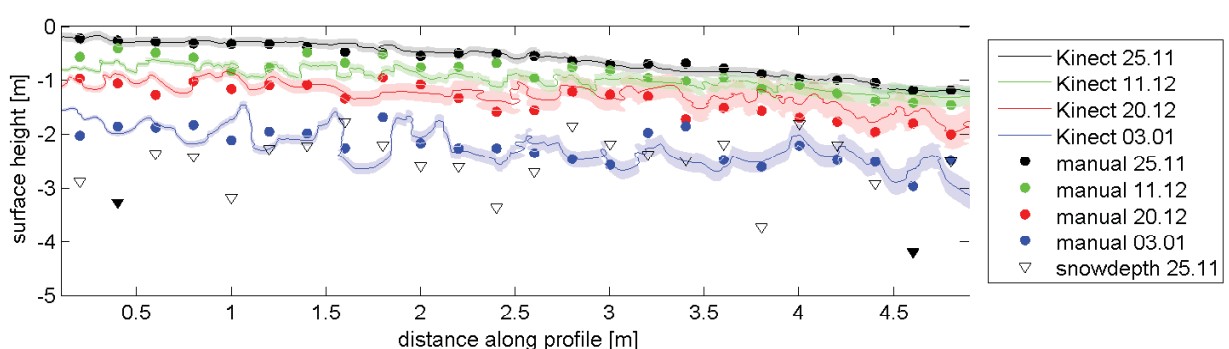