# Peer review of "3D surface properties of glacier penitentes over an ablation season, measured using a Microsoft Xbox Kinect."

_The Cryosphere, 2015_

## Referee Comment (RC1) · R. Naruse (Referee) · 18 Feb 2016

[General Comments]

Snow or ice penitentes are formed on some glaciers or snow patches at high altitudes in the low-latitude regions. Because accesses to these penitent fields with big precision instruments are in general not easy, only overview surveys and qualitative measurements on features or developments of penitentes have been made so far. Then, Nicholson et al. conducted laborious work at Tapado Glacier in the Andes during an austral summer of 2013-14, by measuring geometries of snow penitentes with a laser scanner and an infra-red sensor. These detailed results are valuable for various fields of the cryosphere research, so that this study should be highly evaluated.

However, I have to say that the present paper is not well structured and not sufficiently refined. In other words, the purpose of this paper is unclear, and the manuscript itself is quite long with lots of lengthy paragraphs and sentences (e.g., L74-78, L79-84, L111-114, and others: very hard to read).

I explain this concern more specifically. Abstract should state in principle very concisely, the purpose (in a short sentence), methods, results (findings), interpretations (discussion), and conclusions within one paragraph of 200-300 words. In the present manuscript, the first nine lines (L10-L18) may be moved to Introduction, and the last 12 lines (L27-L39) emphasizes only aerodynamic roughness parameters.

As expressed in the first part of Introduction (L41-L60), it is known that sublimation from the tips of penitentes and concentrated solar radiation in the hollows are essential to the formation of penitentes. On the other hand, turbulent heat flux, which is related with aerodynamic roughness heights, may play negative roles for penitent developments. In Introduction, following the albedo effect (L63-69), roughness parameters are described in detail from L70 to L96.

Thus I guess that the authors' largest interest may be the derivation and properties of roughness parameters. If so, the structure and the way of writing should be significantly modified in order for readers to understand easily the authors' statements.

Since the field measurements were made in an ablation season (of penitentes), typical data for formation of penitentes could not successfully be obtained. However, quantitative information collected on morphologies of penitentes and their changes in time should be precious, since they are typical, peculiar surface features of glaciers.

Issues on the penitent morphology and the aerodynamic roughness are not well harmonized in the present paper. Thus, I suggest now to divide the manuscript into two papers, such as, for example (only for authors' information):

a) "3D surface properties of snow penitentes and their evolutions in an ablation season

2013-14, at Tapado Glacier in the Andes”

[Fig.1, Fig.4, Fig.10, Fig. (meteorological condition), Fig. (heat balance) ]

b) “Aerodynamic roughness parameters over a field of glacier penitentes derived from measurements with a Microsoft Xbox Kinect”

[Fig.1, Fig.2, (Fig.3), Figs. 5, 6, 7, 8, (9) ] The manuscript b) needs to be reviewed by (an) expert(s) on boundary layer micrometeorology.

Renji Naruse (NPO, Japan) Glacier and Cryospheric Environment Research Laboratory

---

## Referee Comment (RC2) · Anonymous Referee #2 · 29 Feb 2016

Recommendation: This paper is suitable for publication after minor revisions.

line 57. Drewry 1970 is missing from the reference list.

line 74. Change Carol to Carroll.

line 80. Fassnacht et al 2009. Do you mean 2009a or 2009b?

line 85. Change "surface" to "survey".

line 100. Fassnacht et al 2010 is not in the reference list.

line 104-105. Change "Mankoff et al." to "Mankoff and Russo". Also on lines 165 and 455.

lines 107-109. This sentence will be easier to read if written with parallel construction. Change to "(i) to perform the first . . . (ii) to perform an examination . . . (iii) to compare the volume . . ."

line 114. Change "mas" to "mass"

line 121. Change "Figure 1b" to "Figure 1"

line 267. Change Brutseart to Brutsaert.

line 357. "strong east-west preferential orientation". But Figures 4e and 4f show aspect orientation north-south, not east-west. You need to define "aspect".

line 382. Change Lettaus to Lettau's

line 401. "at site A". Figure 8 caption mentions only site B, not site A.

line 409. "predominantly from the south-westerly sector". On Figure 9 the wind direction is predominantly SE not SW.

lines 522-525. This is an important result; it should be included in the abstract.

line 674. Breidamerkurjökull. (change h to k)

line 682. Change "roughness of" to "roughness on"

Table 2. Units of windspeed should be m/s. So change the exponent from -2 to -1.

Table 3. This table is not referenced in the text.

Figure 1. What are the units of the tick labels? They should be replaced with latitude and longitude, or else removed.

Figure 2. The labels on the color scale will be easier to read if given in mm instead of m. Then (for example) "0.0000" becomes "0" and "0.0500" becomes "50".

Figure 3 (a,b). Why are the heights negative?

Figure 8. What do the colors mean? What does it mean that the green values are positive but the blue values are at -1m?

[Figure]

---

## Author Comment (AC1) · 25 May 2016

We would like to thank Renji Naruse for the comments on our manuscript and suggested improvements.

The review called for considerable rewriting of the manuscript to improve the clarity of purpose of the study, and we have made every effort to do this, as detailed below. In particular we focused on clearly stating the purpose, and improving the readability, and believe theta the changes detailed below have significantly improved the manuscript.

Reviewer comments are given in green, authors reply in black and revised text sections included in blue italics. The revised manuscript and figure captions are appended to this reply highlighting all changes made, and including updated versions of the figures.
* * *
The purpose of this paper is unclear, and the manuscript itself is quite long with lots of lengthy paragraphs and sentences (e.g., L74-78, L79-84, L111-114, and others: very hard to read).

The aims of the paper have been stated more clearly at the end of the introduction as follows:
"*In this study a Microsoft Xbox Kinect sensor is used as a close-range mobile distance ranger to produce a series of small-scale digital surface models (DSMs). The method of DSM generation is evaluated against standard terrestrial laser scanning and the Kinect-derived DSMs of the penitentes are used to (i) perform the first detailed examination of the morphometry of natural penitentes over the course of an ablation season; (ii) compare the volume change computed from DSM differencing with estimates based on manual measurements of surface lowering and (iii) examine the geometrical roughness properties of the sampled penitentes.*"
The sequencing of the results and discussion has also been reordered to follow the numerical order of these 3 aims.

The manuscript has been re-edited to break long sentences into shorter ones to aid readability. As illustration, the first two examples listed have been changed as follows (the third section has been deleted):
- "*While penitentes are a relatively rare form of linearized surface feature in many glacierized environments, in contrast linear crevasses are widespread, and although the impact of wind direction on roughness and the resultant turbulent heat fluxes is generally not treated in glaciology, penitentes offer a unique test bed for investigating the significance of linearized features on effective surface roughness for various wind directions.*" Now reads: "*While penitentes are a relatively rare form of linearized surface feature in many glacierized environments, linear crevasses are widespread, and penitentes offer a unique test bed for investigating the significance of linearized features on effective surface roughness for various wind directions.*"
- "*In general, the physical roughness of snow and ice surfaces are particularly prone to varying in space and time (e.g. Smeets et al., 1999; Brock et al., 2006; Fassnacht et al., 2009), it is desirable to be able to replace relatively logistically and technologically challenging methods of determining roughness parameters from atmospheric profile or eddy covariance measurements, with methods based on more readily measurable surface terrain properties (e.g. Kondo and Yamazawa, 1986; Munro, 1989; Andreas, 2011), or properties such as radar backscatter that can be derived from spaceborne instruments (e.g. Blumberg and Greeley, 1993).*"

now reads "*As it is logistically challenging to deploy instrumentation to determine roughness parameters from atmospheric profile or eddy covariance measurements on glacier surfaces, efforts have been made to instead use methods based on properties such as radar backscatter (e.g. Blumberg and Greeley, 1993) or more readily measurable surface terrain properties (e.g. Kondo and Yamazawa, 1986; Munro, 1989; Fassnacht et al., 2009a; Andreas, 2011).*"

Abstract should state in principle very concisely, the purpose (in a short sentence), methods, results (findings), interpretations (discussion), and conclusions within one paragraph of 200-300 words. In the present manuscript, the first nine lines (L10-L18) may be moved to Introduction, and the last 12 lines (L27-L39) emphasizes only aerodynamic roughness parameters.

Thanks for this; we fully agree that the initial abstract was not of sufficient brevity, or clarity. However, it remains the case that half of the abstract still refers to surface roughness as this part of the analysis is harder to condense than the morphometrical observations. The abstract has now been significantly reduced in length (from 492 to 364 words) following the suggestions and now reads as follows: "*In this study, the first small-scale digital surface models (DSMs) of natural penitentes on a glacier surface were produced using a Microsoft Xbox Kinect sensor on Tapado Glacier, Chile (30°08'S; 69°55'W). The surfaces produced by the complete processing chain were within the error of standard terrestrial laser scanning techniques, but insufficient overlap between scanned sections that were mosaicked to cover the sampled areas can result in three-dimensional positional errors of up to 0.3 m. Between November 2013 and January 2014 penitentes become fewer, wider, deeper, and the distribution of surface slope angles becomes more skewed to steep faces. Although these morphological changes cannot be captured by manual point measurements, mean surface lowering of the scanned areas was comparable to that derived from manual measurements of penitente surface height at a minimum density of 5 $m^{-1}$ over a 5 m transverse profile. Roughness was computed on the 3D surfaces by applying two previously published geometrical formulae; one for a 3D surface and one for single profiles sampled from the surface. Morphometric analysis shows that skimming flow is persistent over penitentes, providing conditions conducive for the development of a distinct microclimate within the penitente troughs. For each method a range of ways of defining the representative roughness element height was used, and the calculations were done both with and without application of a zero displacement height offset to account for the likelihood of skimming air flow over the closely-spaced penitentes. The computed roughness values are in the order of 0.01-0.10 m during the early part of the ablation season, increasing to 0.10-0.50 m after the end of December, in line with the roughest values previously published for glacier ice. Both the 3D surface and profile methods of computing roughness are strongly dependent on wind direction. However, the two methods contradict each other in that the maximum roughness computed for the 3D surface coincides with airflow across the penitente lineation while maximum roughness computed for sampled profiles coincides with airflow along the penitente lineation. These findings highlight the importance of determining directional roughness and wind direction for strongly aligned surface features and also suggest more work is required to determine appropriate geometrical roughness formulae for linearized features.*"

As expressed in the first part of Introduction (L41-L60), it is known that sublimation from the tips of penitentes and concentrated solar radiation in the hollows are essential to the formation of penitentes. On the other hand, turbulent heat flux, which is related with aerodynamic roughness heights, may play negative roles for penitent developments. In Introduction, following the albedo effect (L63-69), roughness parameters are described in detail from L70 to L96.

The section about methods of determining surface roughness has been moved to the methods section immediately prior to introducing the geometrical approach applied in this work. This makes the introduction more streamlined and is both more in balance with, and leads the reader more smoothly to, the goals of the paper.

I guess that the authors' largest interest may be the derivation and properties of roughness parameters. If so, the structure and the way of writing should be significantly modified in order for readers to understand easily the authors' statements. Issues on the penitent morphology and the aerodynamic roughness are not well harmonized in the present paper. Thus, I suggest now to divide the manuscript into two papers, such as, for example (only for authors' information):

a) "3D surface properties of snow penitentes and their evolutions in an ablation season 2013-14, at Tapado Glacier in the Andes" - [Fig.1, Fig.4, Fig.10, Fig. (meteorological condition), Fig. (heat balance) ]

b) "Aerodynamic roughness parameters over a field of glacier penitentes derived from measurements with a Microsoft Xbox Kinect" - [Fig.1, Fig.2, (Fig.3), Figs. 5, 6, 7, 8, (9) ] The manuscript b) needs to be reviewed by (an) expert(s) on boundary layer micrometeorology.

We prefer not to separate the paper into two, as the surface roughness properties fall under the umbrella of the 'surface properties' covered by the title. Additionally, we are keen to include within this paper at least one way in which the surfaces generated by the Kinect scanning can have valuable scientific applications.

However we have tried to respond to the comment that the component parts needed to be better integrated than they were in the initial submission and accordingly a number of changes were made to improve the relationship between the morphology and the roughness parts of the paper. For example:

- L67: "Measurements of natural penitentes required to examine their morphometry and roughness are rare (e.g. Naruse and Leiva, 1997), and …"
- L75: clearly stated the threefold aims "*… are used to (i) perform the first detailed examination of the morphometry of natural penitentes over the course of an ablation season; (ii) compare the volume change computed from DSM differencing with estimates based on manual measurements of surface lowering and (iii) examine the geometrical roughness properties of the sampled penitente surfaces.*"
- These aims are now tackled consistently in this order throughout the paper.
- In keeping with this order, Figure 10 was moved up the order to become Figure 5.
- L470: "*The changing morphometry of the penitentes alters the geometrical surface roughness as they develop over the ablation season.*"

**3D surface properties of glacier penitentes over an ablation season, measured using a Microsoft Xbox Kinect.**

Lindsey. I. Nicholson[1], Michał. Pętlicki[2,3], Ben. Partan[4], and Shelley. MacDonell[3]

[1] *Institute of Atmospheric and Cryospheric Sciences, University of Innsbruck, Innsbruck, Austria*
[2] *Institute of Geophysics, Polish Academy of Sciences, ul. Księcia Janusza 64, 01-452 Warsaw, Poland*
[3] *Centro de Estudios Avanzados en Zonas Áridas (CEAZA), La Serena, Chile*
[4] *University of Maine, Orono, USA*

*Correspondence to:* L. I. Nicholson (lindsey.nicholson@uibk.ac.at)

**Abstract.** ~~Penitentes are a common feature of snow and ice surfaces in the semi-arid Andes where very low humidity, in conjunction with persistently cold temperatures and sustained high solar radiation favour their development during the ablation season. As penitentes occur in arid, low latitude basins where cryospheric water resources are relatively important to local water supply, and atmospheric water vapor is very low, there is potential value in understanding how penitentes might influence the runoff and atmospheric humidity.~~

~~The complex surface morphology of penitentes makes it difficult to measure the mass loss occurring within them because the (i) spatial distribution of surface lowering within a penitente field is very heterogeneous, and (ii) steep walls and sharp edges of the penitentes limit the line-of-sight view for surveying from fixed positions and (iii) penitentes themselves limit access for manual measurements.we solved these measurement problems by using a Microsoft Xbox Kinect sensor to generatesmall sample areas of snow and icein) between November 2013 and January 2014..However, in our study,studied sites~~
[revised manuscript text omitted]

[Figure]

[Figure]

[Figure]

[Figure]

[Figure]

[Figure]

*A: GPS positions of the base of the marker stakes for sites A and B in UTM region 19S, using the WGS84 datum and ww15mgh geoid, showing combined XY, and XYZ standard deviations (std) are less than 2 cm for all stakes.*

|  | easting | std easting | northing | std northing | elevation | std elevation | XY std [mm] | XYZ std [mm] |
|---|---|---|---|---|---|---|---|---|
| **SA-1** | 410909.704 | 0.004 | 6664147.933 | 0.007 | 4774.568 | 0.015 | 8 | 17 |
| **SA-2** | 410910.615 | 0.006 | 6664143.153 | 0.011 | 4773.496 | 0.008 | 13 | 15 |
| **SA-3** | 410908.618 | 0.004 | 6664142.623 | 0.004 | 4773.375 | 0.013 | 6 | 14 |
| **SA-4** | 410907.751 | 0.004 | 6664147.731 | 0.003 | 4774.518 | 0.015 | 5 | 16 |
| **SA-5** | 410908.046 | 0.004 | 6664145.189 | 0.003 | 4773.988 | 0.017 | 5 | 18 |
| **SB-1** | 410911.808 | 0.005 | 6664156.396 | 0.007 | 4775.352 | 0.014 | 9 | 16 |
| **SB-2** | 410913.034 | 0.004 | 6664154.925 | 0.011 | 4775.278 | 0.012 | 12 | 17 |
| **SB-3** | 410911.426 | 0.003 | 6664153.732 | 0.003 | 4775.314 | 0.011 | 4 | 12 |
| **SB-4** | 410910.228 | 0.003 | 6664155.065 | 0.004 | 4775.464 | 0.011 | 5 | 12 |

*B1: Information on the mesh components and alignment errors for each scanned surface at both glacier sites.*

| | Site A | | | | Site B | | |
|---|---|---|---|---|---|---|---|
| | **25-Nov** | **11-Dec** | **20-Dec** | **03-Jan** | **11-Dec** | **20-Dec** | **03-Jan** |
| **# of meshes used** | 13 | 10 | 13 | 10 | 6 | 6 | 3 |
| **# of arcs used (potential arcs)** | 16(28) | 16(21) | 17(28) | 11(19) | 9 | 11 | 5 |
| **mean error [mm]** | 2.396 | 2.632 | 2.995 | 3.171 | 2.524 | 3.241 | 3.484 |
| **median error [mm]** | 2.172 | 2.541 | 3.112 | 2.945 | 2.414 | 3.310 | 3.285 |
| **90th % error [mm]** | 3.186 | 3.541 | 3.567 | 3.836 | 2.784 | 3.781 | 3.386 |

*B2: Detailed mesh-processing procedure used in this study.*

- All processsesing was carried out in Meshlab unsless otherwise stated
- Pairwise point alignment of the component surface meshes covering each study site
- Applied filter to remove mesh sections (vertices and faces) consisting of < XXX vertices
- Applied filter to remove unreferenced and duplicated vertices
- ICP alignment optimization of the mosaicked component surface meshes using the following parameters:
    - sample number of 1000 for each ICP iteration
    - minimal starting distance for chosen points of 10 mm at the first iteration reducing by 20% on each iteration
    - maximum of 50 iterations were performed
    - using rigid matching so that no stretching or warping of the mesh is permitted
    - export distributed alignment error
- Flattened mosaicked surface meshes into a single layer and remeshed using a Poisson filter with the following paramters:
    - Octreee depth (12)
    - Solver divide (7)
    - number of samples per node (1)
- Meshes were georeferenced with differential GPS measurements in Polywork
- Corner marker stakes, and parts of the mesh representing sensors installed within the sample site were manually removed from the georeferenced surface mesh and the mesh was cropped at the margins
- Triangle numbers were reduced by merging vertices closer than 2.5mm
- Resultant non-manifold features were removed
- Closed holes using a 20mm diameter filter. Inspected boundaries of resultant meshes to confirm that all remaining boundaries are on the edges of the sub-sampled area.
- Cropped horizontal areas to a consistent patch size: A 2 x 3.5m; B 1.5 x 1.5m
- Exported as .OBJ file from which the vertex coordinates and face indices and metadata were extracted for subsequent analysis in Matlab.

*C: Comments and recommendations on the Kinect sampling strategy used in this study.*

- Daylight swamps the signal of the Kinect. Over rock surfaces the Kinect worked perfectly as long as the surface was not in direct sunlight. Over snow and ice the effective range was reduced to about 1m and scanning could only be performed once the sun was below the horizon and was even better after darkness had fallen.
- This study used ReconstructMe as the capture software as it performs real time meashing so that the quality of the surface collected can be assessed at the time of capture. This is an advantage for:
  - observing if return signals had been obtained from the troughs of the penitentes as penetration into very narrow penitente troughs was only achieved over several passes and by re-orientating the sensor to be parallel with the trough.
- The disadvantages of ReconstructMe are that:
  - it does not save the raw depth data
  - it requires a  computer with a  powerful graphics processor as the real time processing is performed at the same 30Hz frequency as the depth-map frame production of the Kinect.
  - the powerful graphics processor tends to be power hungry
- Alternative systems for sampling Kinect data are numerous and growing, and the user must do some up to date research to discover the newest developments, but some existing options are to:
  - use the 'KinectFusion' algorithm (Izadi et al., 2011;Newcombe et al., 2011), implemented in the 'Kinfu' program (part of the Point Cloud Library (PCL); Rusu and Cousins, 2011), which allows one to move the Kinect and scan an area or object, automatically stitching together each frame into one large 3D model, while also capturing raw data.
  - for very large areas, the Kinfu implementation has been extended, named Kintinuous, and used to map paths more than 100m long (Whelan et al., 2012).
- When covering an area larger than 1m$^2$ with a Kinect survey it would be advantageous to have a camera boom mounting for moving the Kinect smoothly over the glacier surface, as this would mean larger areas can be scanned in a single mesh. This would save significant work, and additional error involved in aligning and mosaicking the meshes.
- Ground control point markers which have fixed geometric surfaces with known alignment to x, y, z would have facilitated the alignment and mosaicking the component meshes of each scan. On the basis of this study a marker pole with cubes attached to it at fixed heights and known orientations would be ideal. As the surface lowers and more of the marker stake is revealed additional markers should be added at known distances below the previous marker cube.
- A higher number of ground control points to provide redundancy is advisable as in the case of poorly represented locations for georeferencing step, these could be excluded and the remaining points would still allow successful georeferencing.

---

## Author Comment (AC2) · 25 May 2016

We would like to thank the reviewer for their careful reading of this manuscript. All these minor comments have been remedied as detailed below.

Reviewer comments are given in green, authors reply in black and revised text sections included in blue italics. The revised manuscript and figure captions are appended to this reply highlighting all changes made, and including updated versions of the figures.

line 57. Drewry 1970 is missing from the reference list.

This has now been added.

line 74. Change Carol to Carroll.

Done line 80. Fassnacht et al 2009. Do you mean 2009a or 2009b?

Now specifies Fassnacht et al., 2009b line 85. Change "surface" to "survey".

Done line 100. Fassnacht et al 2010 is not in the reference list.

Sorry, ought to have been 2009a, and now has been corrected.

line 104-105. Change "Mankoff et al." to "Mankoff and Russo". Also on lines 165 and 455.

Corrected in all 3 places lines 107-109. This sentence will be easier to read if written with parallel construction. Change to "(i) to perform the first . . . (ii) to perform an examination . . . (iii) to compare the volume . . ."

Now reads: "*The method of DSM generation is evaluated against standard terrestrial laser scanning, and the Kinect-derived DSMs of the penitentes are used to (i) perform the first detailed examination of the morphometry of natural penitentes over the course of an ablation season; (ii) compare the volume change computed from DSM differencing with estimates based on manual measurements of surface lowering and (iii) examine the geometrical roughness properties of the sampled penitente surfaces.*"

line 114. Change "mas" to "mass"

Done line 121. Change "Figure 1b" to "Figure 1"

Done, and we also reference this figure for the other site features described.

line 267. Change Brutseart to Brutsaert.

Done line 357. "strong east-west preferential orientation". But Figures 4e and 4f show aspect orientation north-south, not east-west. You need to define "aspect".

Now reads: "*The morphometry of the sampled penitentes changed visibly over the measured intervals (Figures 3 and 4). The strong east-west lineation and preferred north and south surface aspect predicted from theory developed early and was maintained throughout study period.*"

line 382. Change Lettaus to Lettau's

Done line 401. "at site A". Figure 8 caption mentions only site B, not site A.

The sentence is not intended to refer to Figure 8. We have added the missing reference to Table 3 here that shows the change in calculated roughness properties over time.

line 409. "predominantly from the south-westerly sector". On Figure 9 the wind direction is predominantly SE not SW.

Correct! This now reads: "*Over the whole study period wind direction is predominantly from the south-easterly and north-westerly sectors, and swings through both extreme wind angles used in the roughness calculations here (Figure 9).*"

lines 522-525. This is an important result; it should be included in the abstract.

Abstract now includes: "*Morphometric analysis shows that skimming flow is persistent over penitentes, providing conditions conducive for the development of a distinct microclimate within the penitente troughs.*", and has been substantially shortened in response to comments from R1.

line 674. Breidamerkurjökull. (change h to k)

Done line 682. Change "roughness of" to "roughness on"

Done

Table 2. Units of windspeed should be m/s. So change the exponent from -2 to -1.

Done

Table 3. This table is not referenced in the text.

Thank you. We now refer to this table as follows: "*Table 3 shows the calculated roughness values at each survey date, revealing that while profile-computed roughness length increases monotonically over time at site B, it reduces over the first period at site A, associated with snowfall during this period.*"

Figure 1. What are the units of the tick labels? They should be replaced with latitude and longitude, or else removed.

They were in UTM and have now been replaced with latitude and longitude

Figure 2. The labels on the color scale will be easier to read if given in mm instead of m. Then (for example) "0.0000" becomes "0" and "0.0500" becomes "50".

This has been done

Figure 3 (a,b). Why are the heights negative?

In Figure 4 the heights are all plotted relative to the reference height of the uppermost stake marking out the horizontal reference for manual measurements of surface lowering. This is described in both the text and the figure caption: "*Figure 4: Summary of the DSM properties through time at site A (left) and B (right). Surface height distribution as a percentage of total surface area, in local coordinates [m] relative to the position of the northern end of ablation frame (a & b). Inset tables show weighted mean mesh elevation, range, surface area and surface area as a function of the horizontal area of the sampled site. Distribution of surface angles as a percentage of total surface area (c & d). Aspect distribution as a percentage of total surface area (e & f).*"

Figure 8. What do the colors mean? What does it mean that the green values are positive but the blue values are at -1m?

The colors refer to dates and these are now labelled on the figure. The height coordinate is relative to the NE corner marker of site A, and the caption now reads: "*Figure 9: Examples of (a) N-S, and (b) E-W oriented cross sections sampled at 0.1 m intervals in local coordinates at site B from which effective surface roughness properties were computed using the methods of Munro (1989, 1999).The surface height coordinate is relative to the NE corner marker of site A.*"

**3D surface properties of glacier penitentes over an ablation season, measured using a Microsoft Xbox Kinect.**

Lindsey. I. Nicholson[1], Michał. Pętlicki[2,3], Ben. Partan[4], and Shelley. MacDonell[3]

[1] *Institute of Atmospheric and Cryospheric Sciences, University of Innsbruck, Innsbruck, Austria*
[2] *Institute of Geophysics, Polish Academy of Sciences, ul. Księcia Janusza 64, 01-452 Warsaw, Poland*
[3] *Centro de Estudios Avanzados en Zonas Áridas (CEAZA), La Serena, Chile*
[4] *University of Maine, Orono, USA*

Correspondence *to:* L. I. Nicholson (lindsey.nicholson@uibk.ac.at)

**Abstract.** ~~Penitentes are a common feature of snow and ice surfaces in the semi-arid Andes where very low humidity, in conjunction with persistently cold temperatures and sustained high solar radiation favour their development during the ablation season. As penitentes occur in arid, low latitude basins where cryospheric water resources are relatively important to local water supply, and atmospheric water vapor is very low, there is potential value in understanding how penitentes might influence the runoff and atmospheric humidity.~~

~~The complex surface morphology of penitentes makes it difficult to measure the mass loss occurring within them because the (i) spatial distribution of surface lowering within a penitente field is very heterogeneous, and (ii) steep walls and sharp edges of the penitentes limit the line-of-sight view for surveying from fixed positions and (iii) penitentes themselves limit access for manual measurements.we solved these measurement problems by using a Microsoft Xbox Kinect sensor to generatesmall sample areas of snow and icein between November 2013 and January 2014..However, in our study,studied sites~~
[revised manuscript text omitted]

[Figure]

[Figure]

[Figure]

[Figure]

[Figure]

[Figure]

*A: GPS positions of the base of the marker stakes for sites A and B in UTM region 19S, using the WGS84 datum and ww15mgh geoid, showing combined XY, and XYZ standard deviations (std) are less than 2 cm for all stakes.*

| | easting | std easting | northing | std northing | elevation | std elevation | XY std [mm] | XYZ std [mm] |
|---|---|---|---|---|---|---|---|---|
| **SA-1** | 410909.704 | 0.004 | 6664147.933 | 0.007 | 4774.568 | 0.015 | 8 | 17 |
| **SA-2** | 410910.615 | 0.006 | 6664143.153 | 0.011 | 4773.496 | 0.008 | 13 | 15 |
| **SA-3** | 410908.618 | 0.004 | 6664142.623 | 0.004 | 4773.375 | 0.013 | 6 | 14 |
| **SA-4** | 410907.751 | 0.004 | 6664147.731 | 0.003 | 4774.518 | 0.015 | 5 | 16 |
| **SA-5** | 410908.046 | 0.004 | 6664145.189 | 0.003 | 4773.988 | 0.017 | 5 | 18 |
| **SB-1** | 410911.808 | 0.005 | 6664156.396 | 0.007 | 4775.352 | 0.014 | 9 | 16 |
| **SB-2** | 410913.034 | 0.004 | 6664154.925 | 0.011 | 4775.278 | 0.012 | 12 | 17 |
| **SB-3** | 410911.426 | 0.003 | 6664153.732 | 0.003 | 4775.314 | 0.011 | 4 | 12 |
| **SB-4** | 410910.228 | 0.003 | 6664155.065 | 0.004 | 4775.464 | 0.011 | 5 | 12 |

*B1: Information on the mesh components and alignment errors for each scanned surface at both glacier sites.*

|  | Site A | | | | Site B | | |
|---|---|---|---|---|---|---|---|
|  | **25-Nov** | **11-Dec** | **20-Dec** | **03-Jan** | **11-Dec** | **20-Dec** | **03-Jan** |
| **# of meshes used** | 13 | 10 | 13 | 10 | 6 | 6 | 3 |
| **# of arcs used (potential arcs)** | 16(28) | 16(21) | 17(28) | 11(19) | 9 | 11 | 5 |
| **mean error [mm]** | 2.396 | 2.632 | 2.995 | 3.171 | 2.524 | 3.241 | 3.484 |
| **median error [mm]** | 2.172 | 2.541 | 3.112 | 2.945 | 2.414 | 3.310 | 3.285 |
| **90th % error [mm]** | 3.186 | 3.541 | 3.567 | 3.836 | 2.784 | 3.781 | 3.386 |

*B2: Detailed mesh-processing procedure used in this study.*

- All processesing was carried out in Meshlab unsless otherwise stated
- Pairwise point alignment of the component surface meshes covering each study site
- Applied filter to remove mesh sections (vertices and faces) consisting of < XXX vertices
- Applied filter to remove unreferenced and duplicated vertices
- ICP alignment optimization of the mosaicked component surface meshes using the following parameters:
    - sample number of 1000 for each ICP iteration
    - minimal starting distance for chosen points of 10 mm at the first iteration reducing by 20% on each iteration
    - maximum of 50 iterations were performed
    - using rigid matching so that no stretching or warping of the mesh is permitted
    - export distributed alignment error
- Flattened mosaicked surface meshes into a single layer and remeshed using a Poisson filter with the following paramters:
    - Octreee depth (12)
    - Solver divide (7)
    - number of samples per node (1)
- Meshes were georeferenced with differential GPS measurements in Polywork
- Corner marker stakes, and parts of the mesh representing sensors installed within the sample site were manually removed from the georeferenced surface mesh and the mesh was cropped at the margins
- Triangle numbers were reduced by merging vertices closer than 2.5mm
- Resultant non-manifold features were removed
- Closed holes using a 20mm diameter filter. Inspected boundaries of resultant meshes to confirm that all remaining boundaries are on the edges of the sub-sampled area.
- Cropped horizontal areas to a consistent patch size: A 2 x 3.5m; B 1.5 x 1.5m
- Exported as .OBJ file from which the vertex coordinates and face indices and metadata were extracted for subsequent analysis in Matlab.

*C: Comments and recommendations on the Kinect sampling strategy used in this study.*

- Daylight swamps the signal of the Kinect. Over rock surfaces the Kinect worked perfectly as long as the surface was not in direct sunlight. Over snow and ice the effective range was reduced to about 1m and scanning could only be performed once the sun was below the horizon and was even better after darkness had fallen.
- This study used ReconstructMe as the capture software as it performs real time meashing so that the quality of the surface collected can be assessed at the time of capture. This is an advantage for:
    - observing if return signals had been obtained from the troughs of the penitentes as penetration into very narrow penitente troughs was only achieved over several passes and by re-orientating the sensor to be parallel with the trough.
- The disadvantages of ReconstructMe are that:
    - it does not save the raw depth data
    - it requires a  computer with a  powerful graphics processor as the real time processing is performed at the same 30Hz frequency as the depth-map frame production of the Kinect.
    - the powerful graphics processor tends to be power hungry
- Alternative systems for sampling Kinect data are numerous and growing, and the user must do some up to date research to discover the newest developments, but some existing options are to:
    - use the 'KinectFusion' algorithm (Izadi et al., 2011;Newcombe et al., 2011), implemented in the 'Kinfu' program (part of the Point Cloud Library (PCL); Rusu and Cousins, 2011), which allows one to move the Kinect and scan an area or object, automatically stitching together each frame into one large 3D model, while also capturing raw data.
    - for very large areas, the Kinfu implementation has been extended, named Kintinuous, and used to map paths more than 100m long (Whelan et al., 2012).
- When covering an area larger than 1m² with a Kinect survey it would be advantageous to have a camera boom mounting for moving the Kinect smoothly over the glacier surface, as this would mean larger areas can be scanned in a single mesh. This would save significant work, and additional error involved in aligning and mosaicking the meshes.
- Ground control point markers which have fixed geometric surfaces with known alignment to x, y, z would have facilitated the alignment and mosaicking the component meshes of each scan. On the basis of this study a marker pole with cubes attached to it at fixed heights and known orientations would be ideal. As the surface lowers and more of the marker stake is revealed additional markers should be added at known distances below the previous marker cube.
- A higher number of ground control points to provide redundancy is advisable as in the case of poorly represented locations for georeferencing step, these could be excluded and the remaining points would still allow successful georeferencing.

---

## Author Response (AR2)

Dear Tobias,

We would like to thank both you and the reviewer for your suggestions to further improve the clarity and readability of this manuscript. We have addressed each point in turn below and in particular have substantially reduced the length of the paper. We hope the changes we have made here meet your requirements for publication.

Lindsey and co-authors

**Comments from the editor**

I feel that the paper can be shortened by ~20% without loss of relevant information.

*The text has been reworked to reduce its overall length (excluding abstract) from 9020 words to 7693 words.*

Abstract/General content:

You nicely write about the general importance of the study, e.g. with respect to understanding glacier changes and water resources. However, in the abstract no information about the results in this respect are given. Be more quantitative and include at least rough information about the mean measured surface lowering (and an estimate for the entire glacier, if possible).

*Here we have added some specific values: "Between November 2013 and January 2014 penitentes become fewer, wider, deeper, and the distribution of surface slope angles becomes more skewed to steep faces. Surface lowering during this core ablation season was in the order of 0.04m day$^{-1}$."*

*In the absence of supporting data, we prefer not to provide an estimate for the whole glacier on the basis of measurements over <10m$^2$.*

L. 14: You mention a potential error of up to 0.3 m due to insufficient overlap. However, it is unclear from this sentence whether this is a specific problem of your study and how the 0.3 m affects the significance of the result.

*This was quoting the single worst alignment to an individual ground control point in our study. This has been clarified and now reads: "The three-dimensional positional error of alignment between the digital surface and ground control points, were on average 0.08m, but in one case reached 0.3 m, due to poor overlap of individual scanned sections comprising the surface."*

*The impact of errors of alignment to the GCPs is included in the calculations of volume change, and in the comparison to the manual measurements.*

L. 25: "in line with the roughest values previously published for glacier ice". This is also unclear.

This now reads: *"The computed roughness values are in the order of 0.01-0.10 m during the early part of the ablation season, increasing to 0.10-0.50 m after the end of December, in line with the largest previously published surface roughness values for glacier ice."*

Methods:

L. 81ff (Section 2.1). The heading is misleading. You include here also information of measurements, but no information about the general climatic conditions (which I expected here).

The heading has been changed to: "Description of field area and measurement setup" to better accommodate the material we wish to cover here.

Additional information regarding the climatic conditions has been included: *"Interannual climate variability is controlled by the El Niño Southern Oscillation (ENSO), such that during El Niño events, higher precipitation and warmer conditions are experienced (Escobar and Aceituno, 1998). Most precipitation is received during the winter (Vuille and Ammann, 1997), however convective storms can cause small precipitation events in the period from December to March (Schotterer et al., 2003). Although the glacier mass balance in the area is highly sensitive to precipitation, warming at elevation over the last 40 years has produced a rise of the glacier equilibrium line al"titude of over 120 m (Carrasco et al., 2008). Annual mean temperature is below freezing and annual mean relative humidity is below 30% (Ginot et al., 1999). The glacier experiences year-round ablation by sublimation, however, melt is only produced during the summer (Sinclair and MacDonell, 2016).*

L. 85: "study areas". This term might be misleading. I understand that you had two test sites, correct? Then write so.

Terminology changed to 'test site' and 'glacier site', and described more explicitly.

L. 116: Include short information about how you corrected for atmospheric conditions and where you got the information about the conditions (probably from the AWS but not clear from the sentence).

Corrections were made using data from a weather station in the glacier forefield, and this is now stated: *"Resulting point clouds were corrected for atmospheric pressure, temperature and humidity using data from a weather station in the glacier forefield, and then trimmed using ILRIS Parser software,…"*

L. 122: Unclear here why the measurements could not be done on the same day.

This was due to a logistical error, and subsequently limited availability of the TLS and Kinect at the same time (TLS *en route* to Antarctica), and is now stated as such.

Section 2.7.This section is hard to read and to understand. It is also too long. Write briefly about the experience from previous studies and then provide a rational for your approach based on these experiences and describe then clearly what you did. It might also be worth to think to move some info about previous studies into the introduction.

The section has been significantly shortened. Superfluous material has been removed, and relevant information moved to the introduction and discussion sections.

L. 270/271: Here you describe the test site and where TLS scans were taken. These sentences fit, hence, better into the methods sections.

The two sentences have been removed as the information was already presented in the method section.

Methods:

L. 284: The differences are not in figure 2b but in the data. Correct the sentence. What do you mean with "holes"? "Data gaps"?

This now reads: *"The difference between the two aligned meshes where overlapping data existed was always < 2 cm (Fig 2b), which is well within the uncertainty of the georeferenced TLS surface model. Larger differences of up to 5 cm, evident in Figure 2b, occur only where there are data gaps in one of the surfaces being compared."*

'Holes' replaced with 'data gaps'.

Section 3.2 About meteorological conditions.
This section is a bit misplaced here. The knowledge about the meteorological conditions and the methods of measurements should be presented earlier and not in the results section.

This section has been removed. Table 2 is referred to in the methods section instead, and further salient points are moved to the discussion.

Additional information regarding the general climatic conditions is presented in the field site description.

Section 3.3 "Aerial scans of penitents surfaces"
The heading fits not well as the first part of this section presents important information about surface lowering and mass loss. This is important and more information could be provided. Tell about the mass loss of the entire glacier based on these measurements here.

This heading has been changed to: *"Morphometric changes and surface lowering"*

L. 313: The sentence and the approach is not really clear to me. How did you calculate the volume change and the surface lowering? This should be described in the methods section and not the results section.

These methods are described in the methods section: *"As the surfaces contain overhanging parts, DSM differencing cannot be performed by simple subtraction. Instead surface lowering was calculated in two ways: Firstly by differencing area-weighted mean surface elevations, and secondly by computing the volume change between scan dates. For the latter approach, volumes for all surfaces were computed relative to a baselevel horizontal reference. Volumes relative to this horizontal reference for upward-facing triangles were computed column-wise, by projecting the area of each triangular face onto the reference surface and using the height coordinate of the triangle centroid as the height dimension for each column. These were summed and volumes for overhanging triangles, calculated in the same way, were subtracted to derive the total volume between the reference surface and each scanned penitente surface. Successive volumes were then subtracted to obtain the volume change over each measurement interval."*
We now refer to area-weighted mean surface height rather than hypsometric mean surface height in the hope that this is clearer to most readers.

L. 322 and elsewhere (e.g. but not limited to L. 345/346): Use uncertainty instead of error. Error refers to the deviation of the truth.

This has been done throughout the text.

L. 343: "compare well". Be more precise.

Now reads: *"This differs from the value calculated from volume change computed from surface meshes consisting of over 1.3 million points and covering an area of 7 $m^2$ by only 28 kg $m^{-2}$, which is within the uncertainty of the two measurement methods."*

L. 364ff and L. 995ff: These lines fit better in the discussion and could shortly be further elaborated.

These lines now open the discussion.

Discussion:

General: The discussion is quite lengthy, hard to follow and should more precise. Shorten and link the sections better to each other. Section 4.1 and 4.3: The surface roughness depends on the morphology. They can maybe be combined.

The discussion has been considerable shortened and re-ordered. However, we kept the morphology and roughness sections separate.

L. 402: "broadly meet" be more precise.

We now say 'are similar to'

L. 408/410: shortwave "radiation".

Corrected

L. 420: Was the modelling done by Lhermitte et al. 2014? This study seems to be quite relevant as it addresses the same glacier. Present this study in the Introduction.

This paper was already referenced in the introduction, but we now highlight its findings as follows: *"Previous studies, based on radiative modelling within idealized penitente surfaces, have investigated the impact of penitentes on the shortwave radiative balance (Corripio and Purves, 2005; Cathles et al., 2014; Lhermitte et al., 2014). The results suggest that penitentes reduce effective albedo by up to 40% compared to flat surfaces and that both shape and penitente size impact the apparent albedo as measured by ground and satellite sensors (Lhermitte, et el., 2014)."*

L. 453ff: These are important results and do not fit in a discussion about methods. I suggest moving to the results section and discuss the implication in the discussion section.

Now in results section 3.3

L. 458: How can you state that these measurements would underestimate the mass loss of the entire glacier? Are mass balance measurements existing?

Accurate mass balance measurements are not available for the period of the study. We therefore remove reference to the glacier-wide mass balance and simply state: *"Assuming that this difference holds true for the whole ablation season of 120 days, point measurements underestimate the seasonal mass loss obtained from the Kinect digital surface models by 86 kg m$^{-2}$."*

L. 488: What kind of glaciers do you refer to?

The highest values are for rough ice in the ablation zone and this now reads: *"These values are in line with values previously published for rough glacier ice (Smeets et al., 1999; Obleitner, 2000)."*

Conclusions:

The conclusions should be more specific. Where should and could the presented methods be used?

We added some specifics regarding potentially useful applications: *"This study demonstrates that the Microsoft Kinect sensor be used successfully at close range over rough snow and ice surfaces under low light conditions, to generate small-scale digital surface models useful for assessing morphometry and surface roughness properties of complex terrain, as well as detailed assessments of spatial variability of*

*surface ablation. The data collected in this study offers the first detailed study of how the geometry of penitentes evolve through time, highlighting the rate of change of surface properties over an ablation season that can serve as a guideline for parameterizing surface properties required for energy and mass balance modelling of penitente surfaces. The method demonstrated here could be useful for investigating glacier surface features such as sastrugi, crevasses or meltwater streams and determining the patterns of surface change associated with such features."*

L. 534f: Here you mention the first time a "number of potentially superior alternative set-up ... and software is now available". This is important information. These alternatives should be shortly mentioned and discussed in the discussion section. Maybe some of your problems can be overcome with other alternatives.

Removed this from the conclusions and instead included the following in the discussion: *"The practical utility of the Kinect on glacier surfaces is limited to small study areas, but integrating local findings with glacier wide TLS or photogrammetric information of surface conditions may offer a means to usefully extrapolate small scale findings to the glacier scale. Surface scanning technology and software is an area of rapid development, and ongoing development of new sensors and airborne platforms may eliminate the challenges of producing high quality depth maps over larger areas using similar technology to the Kinect."*

Supplement:
The Supplementary files should be in a single pdf and only the ply scripts in one .zip fie.

This has been prepared as requested but note that it seems only one document can be uploaded as supplementary material. Hence the zip file containing the data will be provided on request.

**Comments from reviewer**

The purpose of this paper has become clear from the first sentence of Abstract "In this study, the first small-scale digital surface models (DSMs) of small natural penitentes on a glacier surface were produced using a Microsoft Xbox Kinect sensor on Tapado Glacier, Chile (30°08'S; 69°55'W)."

Then one of the major results, though sounding negative for the method developer, may be concluded as that the MXK method does not have a distinct advantage for studies of mass balance or changes of glaciers, judging from "surface lowering was comparable to that derived from manual measurements" L20-30 in Abstract, L600-601 in Discussion, and L733-734 in Conclusion.

This has been highlighted as a key finding in the abstract: *"Although these morphological changes cannot be captured by manual point measurements, a key finding is that mean surface lowering of the scanned areas was comparable to that derived from manual measurements of penitente surface height at a minimum*

*density of 5 m$^{-1}$ over a 5 m transverse profile, indicating that more limited manual measurements adequately capture the mean lowering of the complex surface."*

However, comparisons of the results by MXK with those by TLS and manual measurements are not well explained in the text, especially Section 3.3 and 3.4 are not clear for what the authors intend to state. Also, Figure 5, which shows the result of comparisons, is not well illustrated with very poor captions.

The results and discussion sections comparing the manual and Kinect measurements have been simplified to clarify the comparison of total change over time.

The Figure 5 caption now reads: "Figure 5: Comparison of surface height through time from manual measurements (points) and extracted from the Kinect scans (solid lines ± vertical error) along the horizontal reference (site A, Figure 1). Triangles indicate original snow depth compared to the surface measured on 25/11/13 and solid black triangles indicate locations where snowdepth exceeded the length of the 3 m probe."

If the authors wish to stress much more 'surface properties', that may be surface roughness, presentations of comparisons of surface lowering, meteorological conditions and glaciological issues can be reduced.

The sections have been reduced as suggested by the reviewer, and as outlined in the response to the editor.

Although I recognize that the revised manuscript has been improved much, the paper is still lengthy and not easy to read in some parts. It is recommended to refine the paper further.

The manuscript has been substantially shortened and we hope this now also makes it clearer to read.

---

## Author Response (AR3)

Minor changes made to final submitted manuscript:

References added to Supplement

Supplement parts numbered: S1, S2 Table S1, Table S2. Referred to as such in the text.

Data availability section added that reads: "*Surface meshes used in this study are provided in the supplementary material (naming convention is site_ddmm.ply). Interactive 3D views of surfaces from site B can be seen at: https://sketchfab.com/LindseyNicholson/folders/penitentes-on-glaciar-tapado-chile. Processing scripts are available on request.*"

L490:    changed "*These values are in line with the roughest values previously published for glacier ice.*" to read "*These values are greater than the roughest values previously published for glacier ice, which are < 0.10 m.*" as this was a mistake.

L256:    changed "*and S is the total horizontal area of the surface components above zd.*" to read "*and S is the total horizontal area of the study site.*" as this was a mistake

Added citations of: Smith, M. W., Quincey, D. J., Dixon, T., Bingham, R. G., Carrivick, J. L., Irvine-Flynn, T. D. L., and Rippin, D. M.: Aerodynamic roughness of glacial ice surfaces derived from high-resolution topographic data, J. Geophys. Res.-Earth, 1–19, doi:10.1002/2015JF003759, 2016., as this paper is relevant to our study and was published during the review process of our paper. Citations added at L74, 209, 215, 236 and 499). Reference added to list.